# Sample Smart, Not Hard: Correctness-First Decoding for Better Reasoning in LLMs

**Xueyan Li**[1,2]    **Guinan Su**[2,4]    **Mrinmaya Sachan**[1]    **Jonas Geiping**[2,3,4]
[1]ETH Zurich    [2]Max Planck Institute for Intelligent Systems
[3]ELLIS Institute Tübingen    [4]Tübingen AI Center

## Abstract

Large Language Models (LLMs) are increasingly applied to complex tasks that require extended reasoning. In such settings, models often benefit from diverse chains-of-thought to arrive at multiple candidate solutions. This requires two competing objectives: to inject enough stochasticity to explore multiple reasoning chains, and to ensure sufficient accuracy and quality in each path. Existing works pursue the first objective by increasing exploration at highly uncertain steps with higher temperature or larger candidate token sets, while others improve reliability by rejecting samples with low confidence post-generation, implying that low confidence correlates with low answer quality. These two lines of thought are in conflict, as they conflate different sources of uncertainty. To resolve this, we argue that the decoding rule should be calibrated by *correctness*, not confidence alone. We should sample from tokens with higher estimated correctness, and reduce sampling where expected correctness is low. We propose simple strategies that achieve this goal: **Greedy-Threshold** makes sampling greedy at very low confidence steps. **Calibrated-TopK** and **Calibrated-$\varepsilon$** set truncation threshold based on estimated rank-wise correctness. Together, our findings challenge prevailing heuristics about decoding under uncertainty, showing consistent gains across reasoning benchmarks, with up to 6% improvement in AIME.

## 1 Introduction

Large Language Models (LLMs) are used for a wide range of generation tasks, ranging from open-ended text to structured problem-solving. In many cases, producing more than one candidate output improves not only fluency, but also reliability, since different samples may capture alternative valid continuations (Wang et al., 2023; Lin et al., 2024). This practice highlights a fundamental trade-off: introducing enough randomness to explore multiple options while still ensuring the accuracy and quality of each individual output (Tan et al., 2024; Meister et al., 2024; Shi et al., 2024). Existing works optimize exploration by raising temperatures or enlarging candidate token sets step-by-step (Nguyen et al., 2025; Zhang et al., 2024; Hewitt et al., 2022). These methods assume that *higher entropy is a signal of uncertainty* between multiple valid next steps, warranting broader exploration. In parallel, other works filter after generation, relying on the finding that *low confidence correlates with low answer quality*. Fu et al. (2025) accepts only samples with high token confidence and stops generation when uncertainty spikes. Hallucination detection also makes use of low-confidence segments (Chang et al., 2024).

These two perspectives are in conflict, because they conflate different sources of uncertainty. From a classical perspective, if a probabilistic language model closely approximates the true distribution over next tokens, then high predictive uncertainty indicates that multiple continuations may be valid. In this case, uncertainty reflects *aleatoric variability*, and broader sampling is appropriate. However, if low-confidence positions are instead those where the model is most often wrong, then additional randomness amplifies *epistemic uncertainty*, which is systematic errors arising from the model's lack of knowledge (Yadkori et al., 2024). In such cases, drawing more samples from an unreliable distribution might compound the model's errors.

In this work, we start by analyzing the role of low-probability tokens in reasoning tasks. We observe that increasing exploration at low-confidence steps is indeed a sub-optimal strategy, as a single

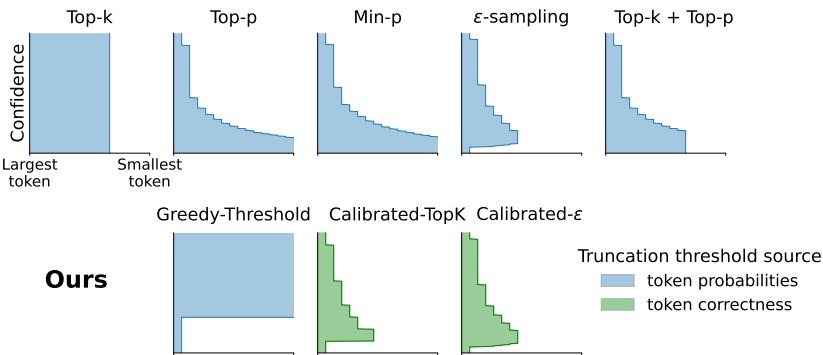

**Figure 1:** Comparison of common and our proposed truncation strategies. Each panel shows which tokens remain available for sampling, with tokens ordered from highest to lowest model-assigned probability (left to right). The $y$-axis represents the max token probability ("confidence"). Our methods explicitly suppress low-confidence tail tokens.

misstep can derail subsequent tokens (Arora et al., 2023). This is especially true for smaller models. Therefore, we propose using the counterintuitive, but simple, **Greedy-Threshold** rule that inverts common sampling heuristics in the literature: when a step's maximum probability falls below a threshold, decoding becomes greedy. Greedy-Threshold can be used in addition to existing samplers and shows consistent gains in reasoning benchmarks, especially for smaller models.

With this we build upon prior work on $\varepsilon$-sampling (Hewitt et al., 2022) that drops every token below $\varepsilon$. Previous work chose very small $\varepsilon$ for machine translation or MBR decoding ($\approx 3 \times 10^{-4} - 9 \times 10^{-4}$ in the original paper) (Jinnai et al., 2024; Finkelstein & Freitag, 2024). We show that for reasoning tasks, larger $\varepsilon$ is safe. This is based on the same conservative principle where less randomness is beneficial where the model is epistemically uncertain. An overview of our methods versus those in literature is shown in Figure 1.

Finally, we unify these perspectives by showing that the *rank-wise correctness* of tokens provides better truncation signals than probabilities alone, and propose a learning-free way to approximate rank-wise calibration. **Calibrated-TopK** sets a truncation threshold at each generation step based on estimated correctness for each confidence bin. **Calibrated-$\varepsilon$** extends upon this by replacing discrete confidence bins with a smooth mapping from probability to correctness. It improves from $\varepsilon$-sampling by making the truncation threshold *data-calibrated*. Our paper makes the following main contributions:

- We find that sampling at low-confidence steps contributes little additional diversity, while increasing the risk of selecting low-correctness tokens that can harm overall performance.
- We verify this empirically by showing that a **Greedy-Threshold** that eliminates unreliable tail tokens alleviates this trend and improves reasoning benchmarks when used in addition to existing samplers such as top-$p$, top-$k$ and min-$p$.
- We introduce a **rank-conditional calibration grid** and derive **Calibrated-TopK** and **Calibrated-$\varepsilon$**, learning-free correctness-aware truncation rules that align exploration with expected correctness and incur negligible inference cost.
- We open-source a unified, composable implementation of common samplers and our methods in one framework. [1].

## 2 WHY WE NEED STRICTER SAMPLING FOR REASONING

Before introducing our samplers, we first examine how confidence relates to accuracy across models and how errors emerge at low-confidence steps. We show that token probabilities provide strong signals of correctness: when the model is uncertain (low maximum probability), expected accuracy decreases regardless of model size, and correctness beyond the top-ranked token drops sharply.

---

[1] https://github.com/xueyan-lii/Sample-Smart-Not-Hard

These observations motivate a clear definition of **confidence**, **rank**, and **calibration**, which we use to formalize stricter sampling rules that suppress error-prone low-probability tokens.

## 2.1 DEFINING RANK-WISE ACCURACY

For a prompt–answer pair, we define the gold answer token sequence as $x_{1:L}$ and the sequence generated at inference as $y_{1:M}$. Let $\mathcal{V}$ denote the vocabulary, $|\mathcal{V}| = V$. At any position $t$, the model outputs a logit vector $z_t \in \mathbb{R}^V$ conditioned on a context $h_t$ which is either the gold prefix $x_{<t}$ during calibration, or the generated prefix $y_{<t}$ during decoding. With temperature $T > 0$, the temperature-scaled categorical distribution over the next token is

$$p_t(j \mid h_t; T) = \frac{\exp\left(z_t(j)/T\right)}{\sum_{v \in \mathcal{V}} \exp\left(z_t(v)/T\right)} \quad \text{for } j \in \mathcal{V}. \tag{1}$$

When $T = 1$ we omit $T$ and write $p_t(j)$. Further, let $p_t^{(1)} \geq p_t^{(2)} \geq \cdots \geq p_t^{(V)}$ denote the probabilities sorted in descending order, and let $\mathrm{rank}_t(j) \in \{1, \ldots, V\}$ be the **rank** of token $j$ at step $t$, then top-$k$ sampling draws from tokens with $\mathrm{rank}_t(j) \leq k$. We define **confidence** as the maximum token probability at each step. $p_{t,\max} \triangleq \max_{j \in \mathcal{V}} p_t(j) = p_t^{(1)}$.

**Confidence bins.** We partition model confidence $(0, 1]$ into $n$ contiguous confidence bins, 10 in this work:

$$\mathcal{B}_m = \left(\frac{m-1}{n}, \frac{m}{n}\right], \qquad m = 1, \ldots, n. \tag{2}$$

Each step $t$ is assigned to exactly one bin via the index $m(t)$ such that $p_{t,\max} \in \mathcal{B}_{m(t)}$.

**Rank-wise probability and correctness.** For each step $t$, the rank-wise probability at rank $r$ is $p_t^{(r)}$. Let $x_t^\star \in \mathcal{V}$ be the ground truth next token under teacher forcing. Let $R < V$ be the maximum rank considered. We define the rank-wise correctness as

$$\mathbb{I}\{\mathrm{rank}_t(x_t^\star) = r\} = \begin{cases} 1, & \text{if the gold token appears at rank } r, \\ 0, & \text{otherwise.} \end{cases} \tag{3}$$

**Calibration Grid**. We can estimate a calibration grid over confidence bins and rank just based on given text sequences which we score by teacher forcing. For each bin–rank pair $(m, r)$, we compute the average probability $\hat{p}_{m,r}$, and correctness $\hat{c}_{m,r}$ within confidence bin $\mathcal{B}_m$:

$$\hat{p}_{m,r} = \mathbb{E}\left[p_t^{(r)} \mid p_{t,\max} \in \mathcal{B}_m\right], \qquad \hat{c}_{m,r} = \mathbb{P}\left[\mathrm{rank}_t(x_t^\star) = r \mid p_{t,\max} \in \mathcal{B}_m\right]. \tag{4}$$

In practice, with $N_m$ steps whose $p_{t,\max} \in \mathcal{B}_m$,

$$\hat{p}_{m,r} = \frac{1}{N_m} \sum_{t:p_{t,\max} \in B_m} p_t^{(r)}, \qquad \hat{c}_{m,r} = \frac{1}{N_m} \sum_{t:\, p_{t,\max} \in \mathcal{B}_m} \mathbb{I}\{\mathrm{rank}_t(x_t^\star) = r\}. \tag{5}$$

An example calibration grid with 5 bins is shown in Figure 2 for visualization. Full calibration grids can be found in Section A.10. These definitions apply to any next-token distribution $p_t$ of a language model. If temperature scaling, or any other logit processing is applied, then the calibration would be calculated based on the final probabilities, as in Equation (1).

**Bin-wise expected accuracy.** The expected accuracy for each confidence bin, i.e., the probability of selecting the correct next token, is given by the average rank-wise probability and correctness:

$$C_m = \sum_{r=1}^{R} \hat{p}_{m,r}\, \hat{c}_{m,r}. \tag{6}$$

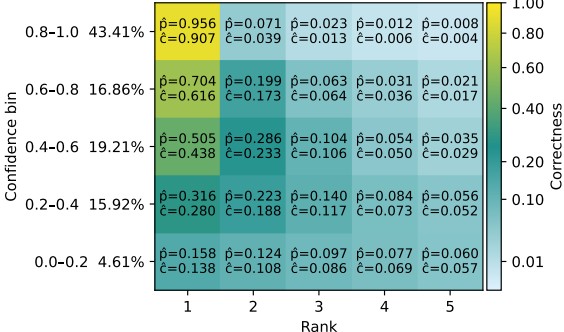

**Figure 2:** Calibration grid of Qwen2.5-1.5B-Instruct with 5 bins shows the average probability $\hat{p}$ and correctness $\hat{c}$ for each confidence-bin and rank. Correctness is notably low in the lower-confidence bins, and decreases as rank increases. Percentages indicate frequency of occurrence of this bin.

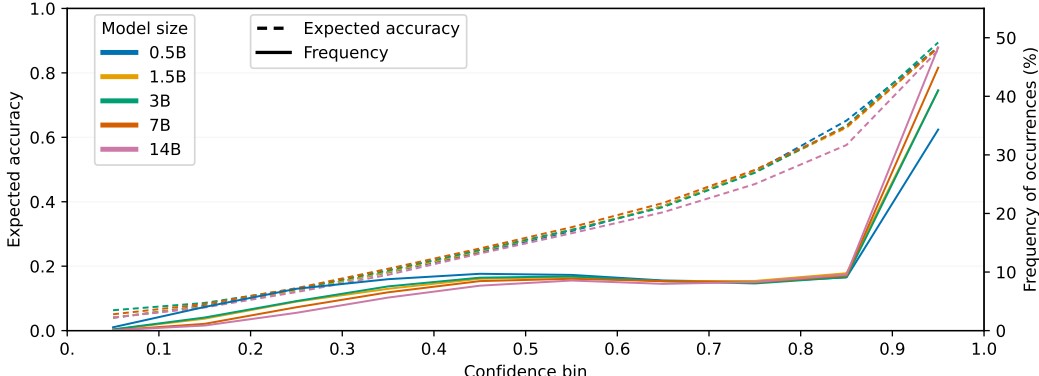

**Figure 3:** Expected accuracy increases with confidence across all model sizes. In the lowest confidence bin, expected accuracy drops regardless of model size. *Frequency* refers to the proportion of decoding steps whose maximum probability falls into each confidence bin. Larger models assign more predictions to the 0.9–1.0 confidence range, where both accuracy and frequency are highest, reflecting stronger benchmark performance. In contrast, smaller models place more probability mass in low-confidence bins, where accuracy is poor.

## 2.2 LOW PROBABILITY SIGNALS LOW CORRECTNESS IN REASONING TASKS

While calibration grids highlight how confidence and correctness align on average, it is less clear how these signals affect full generations. In particular, one might expect that sampling from uncertain positions could encourage exploration which is beneficial over many samples. We test this assumption by analyzing the role of low-probability tokens in self-consistency.

Figure 3 shows that high-confidence predictions occur most frequently, which amplifies diversity simply by providing more opportunities for stochastic sampling. However, this diversity does not necessarily translate into better performance, since rank-wise accuracy drops sharply beyond the top token (Figure 2). In the highest-confidence bin $(0.8, 1.0]$, correctness falls from $0.907$ from rank 1 to only $0.039$ at rank 2. Figure 4 further demonstrates that restricting sampling to the lowest-confidence bin does not yield measurable gains in majority-voted accuracy, while also contributing little to output diversity despite sampling from the full token distribution. This stands in contrast to the assumption that exploration at low-confidence steps is beneficial. Instead, the largest improvements in accuracy arise from sampling in mid-confidence bins $0.3 - 0.6$.

Figure 5 presents two views of why low-confidence positions are dangerous. Figure 5a distinguishes between when a low-probability token is actually chosen (blue) versus when the model is in a low-confidence state regardless of what is sampled (orange). Both conditions harm accuracy as they

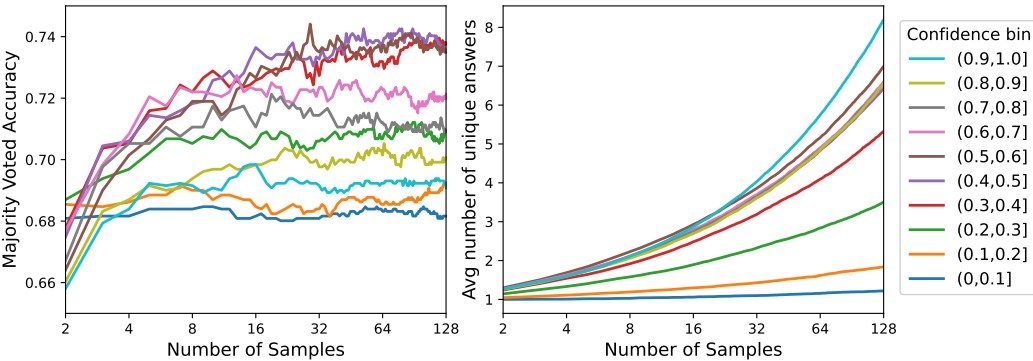

**Figure 4:** Plot of the majority voted accuracy and the number of unique answers as the number of samples increase. Sampling is greedy unless the maximum probability falls into a certain confidence bin, in which case sample from the full token distribution. Sampling at the lowest confidence bin results in no gain in accuracy while contributing little to diversity in terms of number of unique answers.

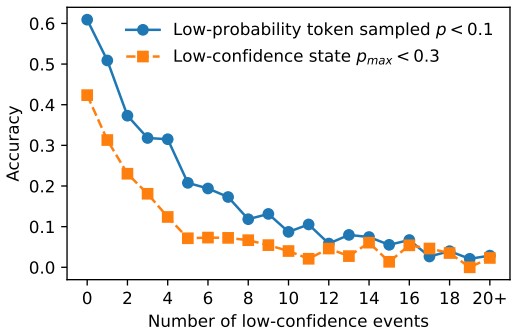 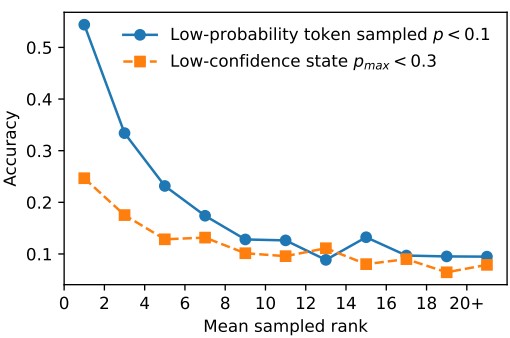

**(a)** Accuracy vs the number of low probability tokens or states sampled in each sequence

**(b)** Accuracy vs mean sampled rank

**Figure 5:** Effect of low-confidence events on sequence-level accuracy. (a) Accuracy decreases both when the model directly samples low-probability tokens ($p < 0.1$, blue) and when it is in a low-confidence state regardless of the sampled token ($p_{\max} < 0.3$, orange). (b) Accuracy also drops as the mean rank of sampled tokens increases, showing that drifting into lower-ranked tokens degrades sequence quality.

accumulate within a sequence. Figure 5b presents a rank perspective: once decoding drifts into higher-ranked tokens, accuracy drops. These findings motivate our conservative truncation rules. By enforcing greedy decoding at low-confidence steps, **Greedy-Threshold** prevents the model from sampling higher-ranked, error-prone tokens and keeps the mean sampled rank low. Similarly, $\varepsilon$-**sampling** blocks low-probability tokens entirely, which naturally caps the rank distribution. In both cases, the methods limit the propagation of errors and preserves sequence-level accuracy.

## 3 How to Calibrate Truncation Samplers

The central idea is to adapt the sampling process in autoregressive language models by filtering out tokens that are likely to be inaccurate, thereby refining the candidate set. Although this might initially seem infeasible, we will show that excluding tokens likely to be incorrect is possible and effective. We call the restricted pool of permissible next tokens the **active set** at each step As a reference, the active set for the simplest truncated sampler, standard top-$k$ sampling, is always the set of the $k$ most likely tokens, i.e. $A_t^{\text{top-k}} = \{\, v \in \mathcal{V} : \mathrm{rank}_t(v) \leq k \,\}$.

**Greedy-Threshold.** We use this sampler to exemplify our claim that sampling less when confidence is low is beneficial. When using Greedy-Threshold, we sample greedily when confidence is below a threshold $p_{GT} \in (0, 1)$, and only the argmax token $v_t^\star \triangleq \arg\max_{v \in \mathcal{V}} p_t(v)$ is accepted. The active set of tokens to sample from is

$$A_t^{\text{GT}} = \begin{cases} \{\, v_t^\star \,\}, & \text{if } p_{t,\max} < p_{GT}, \\ \mathcal{V}, & \text{if } p_{t,\max} \geq p_{GT}. \end{cases} \tag{7}$$

$\varepsilon$-**sampling.** To draw a connection with existing $\varepsilon$-sampling (Hewitt et al., 2022), we recap its definition. This rule only samples from tokens above a threshold $\varepsilon \geq 0$. The active set is

$$A_t^\varepsilon = \{\, v \in \mathcal{V} : p_t(v) \geq \varepsilon \,\}. \tag{8}$$

Note that when the Greedy-Threshold parameter equals the $\varepsilon$-cutoff, i.e. $p_{GT} = \varepsilon$, and the maximum token probability at step $t$ is below this level ($p_{t,\max} < p_{GT}$), both methods fall back to greedy. Motivated by the mismatch between raw probability and correctness, we adopt the same truncation principle but calibrate the cutoff to estimated correctness rather than probability alone.

**Calibrated-TopK.** Recall from the calibration grid (Figure 2) that we can estimate, for each confidence bin and token rank, the expected correctness of a token. This provides a direct way to infer how far down the ranked list of candidates one can safely explore. The idea of Calibrated-TopK is therefore simple. Instead of fixing $k$ in advance, we adaptively set it so that only ranks whose

average correctness is above a threshold are included. In this way, the method truncates exploration to the range of token ranks that are empirically likely to be correct. Given the maximum rank whose correctness is above the threshold $c_{CT} \in (0,1)$ in a calibration grid $\hat{c}_{m,r}$:

$$K_m(c_{CT}) \;=\; \max\{\, r \in \{1, \dots, R\} : \hat{c}_{m,r} \geq c_{CT} \,\} \tag{9}$$

At step $t$ with bin $m(t)$, the active set is defined by the maximum rank

$$A_t^{\mathrm{CT}}(c_{CT}) \;=\; \begin{cases} \{\, v \in \mathcal{V} : \mathrm{rank}_t(v) \leq K_m(c_{CT}) \,\}, & \text{if } K_m(c_{CT}) \geq 1, \\ \{\, v_t^\star \,\}, & \text{if } K_m(c_{CT}) = 0. \end{cases} \tag{10}$$

**Calibrated-$\varepsilon$**  Since Calibrated-TopK sets thresholds based on discrete confidence bins, we are motivated to find a solution that maps probability to correctness in a continuous way. A plot of all $\hat{p}$ and $\hat{c}$ shows a near-linear relationship in log-log coordinates as shown in Figure 6:

$$log_{10}\hat{c} \approx \; A + B \log_{10} \hat{p}$$

We estimate the coefficients by least squares on the calibration grid, fitting a line in log–log space to the pairs $(\hat{p}, \hat{c})$ aggregated over bins and ranks:

$$A, B \;=\; \mathrm{LinearRegression}(\log_{10} \hat{p},\ \log_{10} \hat{c}).$$

Given these coefficients, we instantiate a per-token correctness predictor at decoding step $t$, mapping each candidate token $j \in \mathcal{V}$ with probability $p_t(j)$ to an estimated correctness score $\hat{c}_t(j) \triangleq 10^A p_t(j)^B$. Computationally this is just a single scalar transform, adding negligible overhead to decoding. We then define a correctness threshold $c_\varepsilon \in (0,1)$, the *active set* at step $t$ becomes

$$A_t^{\mathrm{C}\varepsilon}(c_\varepsilon) \;=\; \big\{\, v \in \mathcal{V} \,:\, \hat{c}_t(v) \geq c_\varepsilon \,\big\}.$$

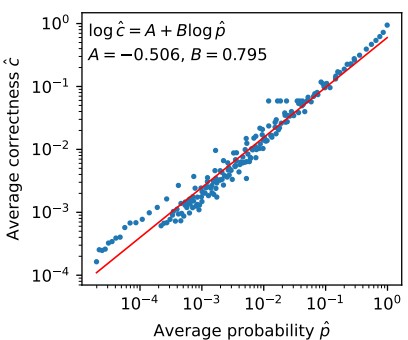

**Figure 6:** A scatter plot of calibration-grid averages $(\hat{p}, \hat{c})$ across confidence bins and ranks. Points concentrate along an approximately linear trend. We fit a least-squares line and use this mapping to predict correctness at inference time for Calibrated-$\varepsilon$.

i.e., we keep exactly those tokens whose *predicted* correctness exceeds the threshold. For all samplers, if no tokens satisfy this condition $A_t = \varnothing$, sample greedily $A_t = \{v_t^\star\}$. In general, we note that all truncated samplers can be used together by taking the intersection of their active sets. Lastly, given any active set $A_t$, sample $x_t \sim p_t'(\cdot)$ from the renormalized distribution

$$p_t'(v) \;=\; \frac{p_t(v)}{\sum_{w \in A_t} p_t(w)} \quad \text{for } v \in A_t. \tag{11}$$

## 4  CORRECTNESS-FIRST SAMPLERS IMPROVE REASONING ABILITIES

We now study whether truncating low-confidence regions during decoding translates into better end-task reasoning. Our focus is on frontier LLMs evaluated on math and general reasoning benchmarks, where sequence-level correctness is the primary objective. We compare the proposed *correctness-first* samplers against standard temperature and probability-based baselines.

### 4.1  EXPERIMENTAL SETTINGS

We evaluate models in the Qwen2.5 (Qwen et al., 2025) and the Llama (Grattafiori et al., 2024) family on short reasoning tasks (GSM8K (Cobbe et al., 2021), MMLU (Wang et al., 2024) and Big-Bench-Hard (Suzgun et al., 2022)), and GPT-OSS (OpenAI et al., 2025) on long reasoning tasks (AIME[2]). We use Greedy-Threshold with $p_{GT} = 0.3$, Calibrated-TopK with $c_{CT} = 0.05$ (over $n = 10$ bins), and Calibrated-$\varepsilon$ with $c_\varepsilon = 0.05$. We adopt higher threshold $\varepsilon = 0.05$ than typically reported to emphasize the impact of truncating low-probability tokens. We provide ablation on threshold value selection in Section A.4. Calibration is performed on the training split of each benchmark. We additionally perform cross-domain calibration using alpaca-gpt4-en (Peng et al., 2023) For comparability with temperature-based samplers, we use $T = 1$ unless otherwise noted. Further implementation details, ablations with threshold values, temperatures and calibration datasets are provided in Section A.1.

---

[2] https://huggingface.co/datasets/math-ai/aime25

**Table 1:** Majority voted (maj@k) results on GSM8K, MMLU-Pro, and Big-Bench-Hard for Qwen2.5-0.5B-Instruct. Calibrated samplers achieve the largest performance gain from no restrictions baseline.

| Method | GSM8K | | | MMLU-Pro | | | Big-Bench-Hard | | |
|---|---|---|---|---|---|---|---|---|---|
| | maj@8 | maj@16 | maj@32 | maj@8 | maj@16 | maj@32 | maj@8 | maj@16 | maj@32 |
| No restrictions | 30.2 | 35.2 | 38.6 | 16.4 | 17.0 | 17.3 | 17.9 | 17.0 | 16.2 |
| top-$k$ | 32.6 | 38.7 | 41.9 | 16.8 | 17.5 | 18.0 | 22.0 | 21.7 | 21.5 |
| top-$p$ | 35.5 | 40.8 | 43.6 | 16.8 | 17.5 | 18.1 | 25.5 | 25.9 | 25.9 |
| min-$p$ | 38.7 | 43.1 | 46.6 | 17.7 | 18.2 | 18.6 | **30.6** | **31.5** | 31.7 |
| EDT | 40.2 | 44.1 | 46.7 | 17.7 | 18.1 | 18.4 | 30.5 | 31.1 | 31.7 |
| $\eta$-sampling | 31.6 | 37.2 | 41.0 | 16.5 | 17.3 | 17.9 | 20.6 | 20.3 | 19.6 |
| $\varepsilon$-sampling | 39.2 | 44.3 | 46.7 | 17.5 | 18.1 | 18.3 | 30.4 | 31.1 | 31.6 |
| **Greedy-Threshold** | 31.2 | 37.0 | 40.6 | 16.9 | 17.8 | 18.2 | 20.8 | 20.3 | 19.4 |
| **Calibrated-TopK** | 39.3 | **44.5** | **47.1** | 17.9 | 18.3 | **18.7** | 30.4 | 31.1 | 31.6 |
| **Calibrated-$\varepsilon$** | **40.8** | 44.3 | **47.1** | **18.9** | **18.4** | 18.6 | **30.6** | **31.5** | **32.0** |

**Baselines** We compare our methods against several widely used sampling strategies. Top-$k$ ($k = 10$) (Fan et al., 2018). Min-$p$ ($p = 0.1$) (Nguyen et al., 2025). Top-$p$ ($p = 0.95$) (Holtzman et al., 2020). EDT (Zhang et al., 2024) with $N = 0.8$, $\vartheta = 1$, $T_0 = 0.7$. These parameters are selected from the original paper after small parameter search experiments to determine reasonable values. $\varepsilon$-sampling (Hewitt et al., 2022) with higher $\varepsilon = 0.05$ than recommended. $\eta$-sampling (Hewitt et al., 2022) with the original recommended value $\eta = 0.0009$.

## 4.2 CALIBRATED TRUNCATION INCREASES MODEL PERFORMANCE.

We evaluate the effectiveness of our proposed samplers across benchmarks. Table 1 shows that **Calibrated-$\varepsilon$** and **Calibrated-TopK** achieve the largest improvement overall, showing rank-wise correctness is an effective truncation signal. **Greedy-Threshold** activates only when the max-probability token falls below 0.3, an infrequent but high-risk regime (see Figure 2). Despite its low activation rate, this condition occurs often enough for Greedy-Threshold to yield measurable benefits. $\varepsilon$-sampling (fixed threshold) performs on par with min-$p$ and EDT, supporting the idea that simply removing tail tokens helps by shaping the cutoff to where correctness actually drops. All calibrated samplers add negligible runtime overhead at decoding, since they only require a 2-parameter table lookup or a single vector operation over the vocabulary.

## 4.3 EXISTING SAMPLERS BENEFIT FROM GREEDY-THRESHOLD

To test whether halting sampling at low-confidence steps is beneficial, we apply Greedy-Threshold on top of existing samplers that otherwise increase exploration at such positions. This preserves their original behavior when confidence is above 0.3, but forces greedy decoding when confidence falls below this threshold. Table 2 shows that Greedy-Threshold improves performance in this setting, especially for smaller models. When no gains are observed, results remain comparable to the baseline, indicating that it does not degrade performance.

## 4.4 SCALING TO ADVANCED REASONING MODELS

We evaluate our proposed methods on reasoning-oriented "thinking" model GPT-OSS-20B (OpenAI et al., 2025) and a challenging mathematics benchmark AIME that demands multi-step derivations. Thinking models differ from standard LMs in that they generate long sequences of intermediate tokens, making calibration on short, instruction-style datasets less representative of their actual behavior. To capture our central idea of filtering out low-correctness tokens in this setting, we apply $\varepsilon$-sampling with a relatively high cutoff ($\varepsilon = 0.05$). Our results show substantial gains. GPT-OSS-20B benefits from both Greedy-Threshold and $\varepsilon$-sampling. Output diversity is reduced, but the effect is minimal. Over 32 samples, the number of unique answers decreases from 14.1 to 13.3 (roughly 1–2 fewer unique answers out of 14). This small drop coincides with higher maj@k and pass@k,

**Table 2:** Majority voted results on GSM8K. In addition to existing sampling conditions, Greedy-Threshold $p_{GT} = 0.3$ is applied. Greedy-Threshold improves majority voting performance, especially in models with lower starting accuracy. Statistically significant differences ($p < 0.05$) marked in **bold**.

| Method | Qwen2.5-0.5B-Instruct | | | Qwen2.5-1.5B-Instruct | | | Qwen2.5-3B-Instruct | | |
|---|---|---|---|---|---|---|---|---|---|
| | maj@8 | maj@16 | maj@32 | maj@8 | maj@16 | maj@32 | maj@8 | maj@16 | maj@32 |
| Baseline $T=1$ | 30.2 | 35.2 | 38.6 | 68.4 | 71.5 | 73.1 | 79.3 | 80.6 | 81.1 |
| + Greedy-Threshold | +1.0 | +1.8 | +2.0 | +1.7 | +2.1 | **+2.4** | +0.1 | +0.2 | -0.1 |
| top-$k$ | 32.6 | 38.7 | 41.9 | 68.5 | 72.6 | 73.9 | 79.0 | 80.3 | 81.0 |
| + Greedy-Threshold | **+1.1** | +0.5 | +1.1 | **+2.6** | **+2.4** | **+2.8** | **+1.0** | +0.5 | -0.2 |
| top-$p$ | 35.5 | 40.8 | 43.6 | 71.1 | 74.1 | 75.9 | 79.5 | 80.5 | 80.8 |
| + Greedy-Threshold | +0.9 | +0.4 | +1.3 | +1.5 | +1.8 | **+1.8** | 0.0 | 0.0 | +0.4 |
| min-$p$ | 38.7 | 43.1 | 46.6 | 73.3 | 75.6 | 76.6 | 80.0 | 80.4 | 81.2 |
| + Greedy-Threshold | +1.4 | +0.8 | +0.6 | +1.3 | +1.2 | +1.5 | -0.2 | +0.2 | +0.1 |
| EDT | 40.2 | 44.7 | 46.8 | 74.9 | 76.6 | 78.9 | 79.5 | 80.5 | 80.9 |
| + Greedy-Threshold | +0.2 | -0.3 | +0.1 | -0.2 | 0.0 | -0.1 | +0.1 | +0.1 | +0.1 |
| $\eta$-sampling | 31.6 | 37.2 | 41.0 | 69.0 | 72.4 | 74.2 | 78.8 | 80.1 | 81.0 |
| + Greedy-Threshold | **+2.4** | **+1.8** | **+1.7** | +1.7 | **+2.7** | **+2.7** | **+0.5** | +0.3 | +0.1 |

consistent with our goal: we do not value diverse wrong answers. For reasoning tasks with single correct solutions, correctness is more important than diversity. Expanding exploration does not help when early steps are error-amplifying. By steering decoding away from low-correctness regions, our methods increase the fraction of valid solutions by up to 6.5% and improve overall answer quality.

## 5 DISCUSSION

**Why does Greedy-Threshold work?** Our results suggest that in reasoning tasks, in spite of popular intuitions, the positions with low confidence are not *branch points among many valid continuations*, but error-amplifying states. Two pieces of evidence support this claim: rank-wise correctness decreases beyond the top token (Figure 2), and performance degrades once low-probability tokens are sampled (Figure 5). Greedy-Threshold chooses a safe token where both risks are highest, and potentially prevent subsequent error. It is a targeted suppression of low-correctness steps.

**Reordering uncertainties as epistemic first, aleatoric second**. Common existing decoding strategies assume high entropy means aleatoric variability (many valid tokens) and sample more. Our results imply the opposite might be true in reasoning tasks with closed-form answers. High entropy

**Table 3:** The result of AIME24 and AIME25 on GPT-OSS-20B with thinking mode enabled. "Unique Answers" is the number of unique answers over all 32 samples. "Overall Correct" is the overall proportion of correct answers. Best result is in **bold**. Statistically significant difference ($p < 0.05$) is in *italics*.

| Model / Method | Maj@k | | | | Pass@k | | | | Unique Answers | Overall Correct |
|---|---|---|---|---|---|---|---|---|---|---|
| | 4 | 8 | 16 | 32 | 4 | 8 | 16 | 32 | | |
| **AIME25** | | | | | | | | | | |
| Baseline | **75.4** | 84.4 | 87.8 | 90.0 | 85.6 | 90.0 | 91.1 | 92.2 | 13.6 | 56.1 |
| Greedy-Threshold | 71.1 | **87.8** | 88.9 | **91.1** | 85.6 | **91.1** | 93.3 | 94.4 | 12.0 | **59.9** |
| $\varepsilon$-sampling | 68.9 | 85.4 | **91.1** | 90.0 | **86.7** | 90.0 | **93.3** | *95.6* | 13.6 | 56.1 |
| **AIME24** | | | | | | | | | | |
| Baseline | 71.4 | 83.3 | 88.7 | **92.6** | 83.4 | 90.7 | 92.0 | 93.3 | 15.1 | 48.7 |
| Greedy-Threshold | **77.3** | **88.0** | **91.3** | **92.6** | 86.7 | **92.6** | **93.3** | **94.0** | *13.3* | **55.2** |
| $\varepsilon$-sampling | **77.3** | *87.3* | 90.0 | 91.3 | **89.3** | 90.0 | 91.3 | 92.6 | *13.7* | *54.9* |

often reflects *epistemic uncertainty* which is a systematic lack of knowledge, especially in smaller models. When the distribution is wrong, sampling more from it does not benefit correctness. In our calibration-based methods, *correctness* is the focus. We increase sampling where expected correctness is high and shrink it where the model lacks fundamental understanding. This perspective explains why stricter truncation (higher $\varepsilon$, lower rank caps in top-$k$) consistently helps in reasoning. Randomness is less valuable in low-confidence regions where epistemic error dominates.

**Robustness to cross-domain calibration data and fit quality.** In-domain calibration dataset is not always available, we want to ensure our calibrated samplers are robust to domain shifts. We test calibration using a general purpose instruction tuning dataset `alpaca-gpt4-en` and observe similar performance as in-domain calibration on GSM8K-training (Section A.5). This suggests that the confidence to correctness map transfers generally when the format matches (instruction to answer) and that out-of-domain calibration datasets can be used when in-domain dataset is not available.

## 6 RELATED WORK

Literature on decoding for LLMs largely follows two directions: (i) *sampling more* to increase diversity of generations, and (ii) *sampling less* to increase accuracy and stability. Our work focuses on reconciling these perspectives for reasoning tasks.

**Removing tail-end tokens**. Classical truncation methods such as top-$k$ (Fan et al., 2018) and top-$p$ (nucleus) sampling (Holtzman et al., 2020) reduce tail risk by discarding low-probability tokens. Temperature scaling (Guo et al., 2017), often with lower temperatures for math and reasoning tasks, has a similar intuition: sharpening the distribution so that low-probability tokens are rarely sampled. These classic methods are often used in combination (Yang et al., 2025). More recently, locally typical sampling (Meister et al., 2025) restricts to tokens whose information content is close to the local entropy. REAL sampling (Chang et al., 2024) adaptively reduces top-$p$ when hallucination risk is high. Our results on reasoning tasks support this risk-aware trend: sampling in high-uncertainty steps introduces catastrophic errors, while removing tail end low probability tokens is safer.

**Selection after sampling.** Another line of work improves reliability *after* generation. Self-certainty (Kang et al., 2025) and DeepConf (Fu et al., 2025) re-weigh or filter generations using confidence signals. The open-source effort Entropix (XJDR, 2024) pauses or resamples at high-entropy steps. These methods implicitly acknowledge that low-confidence steps are strongly correlated with low correctness. Our contribution is orthogonal. We intervene during decoding to prevent low-confidence tokens from being sampled in the first place, so that downstream majority voting operates on stronger candidates.

**Adaptive, uncertainty-aware decoding.** A complementary set of methods dynamically adjust sampling based on estimated uncertainty. Entropy-dependent temperature (EDT) (Zhang et al., 2024) increases temperature as entropy grows. "Hot or Cold" decoding (Zhu et al., 2023) applies higher temperature only to the first token in code generation. Adaptive Decoding (Zhu et al., 2024) and Adaptive Contrastive Search (Garces Arias et al., 2024) adjust candidate sets or penalties step-by-step. Min-$p$ (Nguyen et al., 2025) scales truncation by the maximum token probability, enlarging candidate sets when uncertainty is high, which is shown to benefit creative text generation, although a subsequent study criticized its effectiveness (Schaeffer et al., 2025). Our study provides an explanation why high uncertainty correlates with lower accuracy while providing limited diversity.

**Assessing model calibration**. Calibration for LLMs is typically assessed with reliability diagrams and scalar errors such as ECE error on top-1 labels (Guo et al., 2017), or on a sequence level (Huang et al., 2024; Stengel-Eskin & Durme, 2023). Full-ECE (Liu et al., 2024) extends beyond top-1 by evaluating calibration over the entire token distribution, but it does not condition on token *rank*. We introduce a confidence and rank calibrated method that informs correctness-aware truncation.

## 7 CONCLUSION

**Future work.** This work can be extended to tasks beyond math and reasoning, such as open-ended and creative tasks, to characterize when diversity is the priority while ensuring correctness. There is also the potential to conduct online calibration with on-the-fly recalibration depending on the exact

question or task. Lastly, one can study how calibration evolve with model size, post-training method and data regime.

**Conclusion.** In this work, we re-examined decoding under uncertainty for reasoning tasks and argue for a *correctness-first* perspective. By visualizing a novel rank-wise calibration grid, we present evidence on a **token level** that in low-confidence bins, all tokens have low expected correctness. On a **sequence level**, accuracy declines with more low-probability tokens and with higher ranks sampled. On a **dataset level**, Greedy-Threshold, Calibrated-TopK, and Calibrated-$\varepsilon$ raise performance in reasoning benchmarks by allocating randomness only where expected error is low. These methods are consistent and compute-efficient, making them practical for inference use. We encourage future work to consider uncertainty as a *risk* signal to truncate, rather than a signal to *explore*.

## REPRODUCIBILITY STATEMENT

Implementation of our proposed sampling strategies, model behavior analysis and calibration measurements are released with our paper and hosted in our public repository. Details of evaluation and calibration, including model settings, hyperparameters, and prompting formats are documented in Section A.1. All benchmarks are openly available, and we provide complete code for running the benchmarks.

## LLM USE

We used large language models (LLMs) only for light editorial assistance (e.g., grammar, spelling, phrasing), simple LaTeX formatting, and typo checks. LLMs were also used to draft boilerplate code (e.g., code scaffolding, argument parsers) and to plot high-level diagrams. All such outputs were reviewed, and validated by the authors before inclusion. LLMs were not used to write substantive sections of the paper, design experiments or analyze results. All technical content, experiments, analyses, and conclusions were created and verified by the authors.

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

## A APPENDIX

### A.1 DATASETS AND PARAMETERS

We provide detailed experiment setups and parameters for reproducibility. We evaluate on reasoning and math-focused benchmarks commonly used to assess chain-of-thought robustness. We follow the community-standard evaluation scripts to ensure comparability across papers and release the detailed configuration files in our public GitHub repository.

**Datasets**

- GSM8K (Cobbe et al., 2021): with 5-shot chain-of-thought prompting. The train split is used to construct in-distribution (ID) calibration grid.
- MBPP (Austin et al., 2021): ID calibration grids are built using the training split, with evaluation on the test set. Results are reported in Section A.6
- Big-Bench-Hard (Suzgun et al., 2022): we report unweighted average accuracy across all categories. Since this benchmark does not have a train split, we use alpaca-gpt4-en for calibration.
- MMLU-Pro (Wang et al., 2024): we report unweighted average accuracy across all categories, using the validation split for ID calibration.
- Alpaca-gpt4-en (Peng et al., 2023) [3]: used as an out-of-distribution (OOD) dataset for calibration. This OOD calibration is applied once to derive a general Calibrated-TopK setting, which can then be used across tasks. In contrast, ID calibration offers more precise task-specific signals, but may not always be feasible for new domains. Effect of ID vs OOD calibration dataset is compared in Section A.5
- AIME25[4] and AIME24[5]: used to evaluate "thinking" models with extended reasoning traces.

For all benchmarks, performance is measured as the average maj@k or pass@k across three runs. Popular Python package `lm_eval`[6] is used to evaluate all tasks. `vLLM`[7] is used to run large scale generation for benchmarking. Calibration is done using code bootstrapped to `torchtune`[8].

**Calibration Setup** For all calibration procedures, the input question is masked, and only correctness with respect to ground-truth answers is considered. Unless otherwise specified, we set Greedy-Threshold $p_{GT} = 0.3$, $c_{CT} = 0.05$ for Calibrated-TopK and $c_\varepsilon = 0.05$ for Calibrated-$\varepsilon$. The maximum number of ranks considered is $R = 20$.

**Baselines** We compare our methods against several widely used sampling strategies:

- Top-$k$ ($k = 10$) (Fan et al., 2018)
- Min-$p$ ($p = 0.1$) (Nguyen et al., 2025)
- Top-$p$ ($p = 0.95$) (Holtzman et al., 2020)
- EDT (Zhang et al., 2024) with $N = 0.8$, $\vartheta = 1$, $T_0 = 0.7$. These parameters are selected from the original paper after small parameter search experiments to determine reasonable values.
- $\varepsilon$-sampling (Hewitt et al., 2022) with higher $\varepsilon = 0.05$ than recommended.
- $\eta$-sampling (Hewitt et al., 2022) with the original recommended value $\eta = 0.0009$.

**Long-Form Reasoning** To assess robustness on extended reasoning chains, we test on AIME25 and AIME24 using GPT-OSS's recommended setting ($T = 1.0$), averaged across three runs. Additionally, we restrict baseline comparisons to Greedy-Threshold with $p_{GT} = 0.3$ and $\varepsilon$-sampling with $\varepsilon = 0.05$. As thinking models exhibit substantially different behaviors from standard models,

---

[3]https://huggingface.co/datasets/llamafactory/alpaca_gpt4_en
[4]https://huggingface.co/datasets/math-ai/aime25
[5]https://huggingface.co/datasets/math-ai/aime24
[6]https://github.com/EleutherAI/lm-evaluation-harness
[7]https://github.com/vllm-project/vllm
[8]https://github.com/pytorch/torchtune

using question–answer pairs with short ground-truth answers would produce misleading calibration values.

**Temperature Sensitivity**  To ensure fairness against temperature-based samplers, all main results are reported with $T = 1.0$. Since math and coding tasks often benefit from lower sampling temperatures, we additionally evaluate with $T = 0.6$. In this setting, calibration grids are recomputed using scaled logits (Equation (1)), and thresholds ($p_{GT}$, $\eta$ and $c_{CT}$) are adjusted accordingly. Detailed hyperparameter values and results are provided in Section A.8.

## A.2  PARAMETERS FOR FIGURES

Figure 2 uses 5 confidence bins with 0.2 increments on Qwen2.5-0.5B-Instruct, using GSM8K train dataset.

Figure 3 Models are in the Qwen2.5 instruct family. Expected accuracy is calculated from the top $R = 20$ ranks at each step. Dataset used is `alpaca-gpt4-en` with questions masked, so correctness is only calculated from answers.

Figure 4 Uses Qwen2.5-1.5B-Instruct on GSM8K test set with $T = 1$ and no other sampling conditions. Each final numerical answer is extracted, excluding the reasoning chains. Only different final numerical answers are counted. Different reasoning chains that arrive at the same final answer will count as the same answer. The number of unique answers is averaged over all questions at each number of samples. If no valid final answer can be extracted, the final answer becomes `null`. This means all reasoning chains without a valid final answer will count as one unique answer.

Figure 5 uses Qwen2.5-0.5B-Instruct on GSM8K test set. All reasoning chains up to and including the final numerical answers are considered. Sometimes the model continues to generate the next in-context question after answering the current question. All subsequent generations are excluded.

Figure 6 uses Qwen2.5-0.5B-Instruct on GSM8K train set.

## A.3  EXAMPLE OF LOW-CONFIDENCE CASES

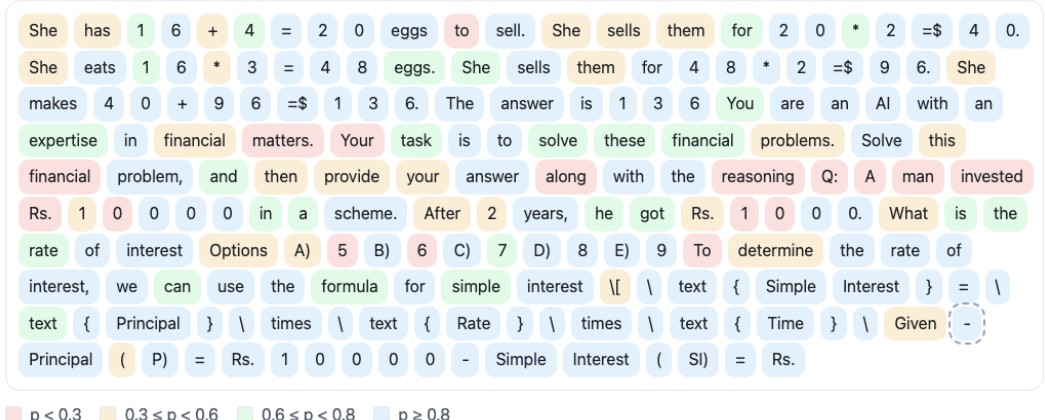

p < 0.3    0.3 ≤ p < 0.6    0.6 ≤ p < 0.8    p ≥ 0.8

**Figure 7:** Example of an answer for GSM8K by Qwen2.5-0.5B-Instruct under greedy generation. The lowest confidence typically do not occur at the start of a sentence.

We illustrate a simple case of where low-confidence tokens arise. As shown in Figure 7, the beginning of a sentence where aleatoric variability is expected, typically exhibits moderate confidence ($p \geq 0.3$). In contrast, very low-confidence tokens ($p < 0.3$) are rarely observed while the model is still producing a coherent initial sentence. Over-sampling at this stage risks introducing irrelevant tokens that derail the generation. Once the model finishes answering the question and shifts to producing the next in-context example, however, both aleatoric variability and epistemic uncertainty increase, and the model's confidence drops substantially.

## A.4 HYPERPARAMETER SEARCH ON PROPOSED METHODS

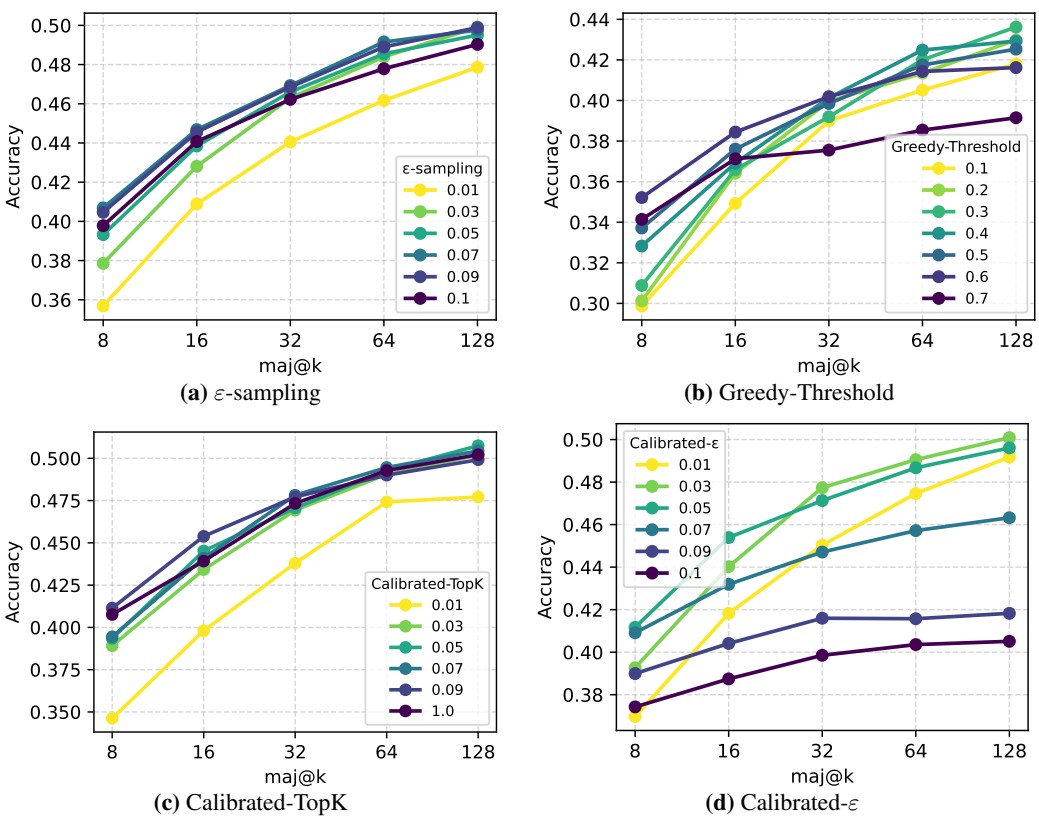

**Figure 8:** Values of each threshold $\varepsilon$, $p_{GT}$, $c_{CT}$ and $c_\varepsilon$ are varied to study their effect on self-consistency performance. Higher number of samples generally benefit from smaller truncation thresholds.

We conduct parameter search for each of our main methods to find the optimal $\varepsilon$, $p_{GT}$, $c_{CT}$ and $c_\varepsilon$ values. Figure 8a confirms our hypothesis that much larger value than suggested in the original $\varepsilon$-sampling paper is beneficial. $\varepsilon$ is $0.0009$ in the original paper (Hewitt et al., 2022). We find that $\varepsilon$ performs best around $0.07 - 0.09$ regardless of the number of samples. For Greedy-Threshold, lower threshold benefit more at higher number of samples. The optimal $p_{GT}$ is 0.3 at 128 samples and 0.6 at 32 samples. Calibrated-TopK yields the best performance at $c_{CT}$ $0.05 - 0.09$ range. Calibrated-$\varepsilon$ shows the strongest gains at $c_\varepsilon = 0.03$.

In general, larger sample sizes benefit from smaller truncation thresholds. This is intuitive: looser thresholds retain more candidate tokens, promoting diversity that enables the model to explore multiple reasoning paths and recover the correct answer often enough to dominate in majority voting. Figure 8 illustrates that optimal threshold selection is inherently sample-size dependent, reflecting the complex trade-off between accuracy and diversity.

In addition, we test various binning methods, including different number of bins and bin widths. Even bins (default) use fixed, uniform bin widths over confidence. Quantile bins adapt their widths so that each bin contains approximately the same number of tokens. This tests whether calibration quality depends on uniform spacing or sample-balanced partitioning. Since high confidence steps are a lot more common, quantile bins would fit high confidence steps more. This results in different linear value fit compared to even bins and worse performance as shown in Table 4. Thus, it is not adviced to use quantile bins. Using different number of evenly spaced bins results in similar performance. We do not overtune the number of bins to avoid overfitting on certain dataset and model combinations.

**Table 4:** Effect of bin size and binning strategy on Qwen2.5-0.5B-Instruct performance on GSM8K. Even bins (default) use fixed, uniform bin widths over confidence. Quantile bins adapt their widths so that each bin contains approximately the same number of confidence samples. $n$ indicates the number of bins. $A$ and $B$ are the bias and gradient for log-log linear fit.

| Bins | A | B | maj@8 | maj@16 | maj@32 |
|---|---|---|---|---|---|
| $n = 5$ even | -0.574 | 0.791 | 41.2 | 45.4 | 47.6 |
| $n = 5$ quantile | -2.290 | 0.435 | 33.9 | 34.1 | 34.0 |
| $10 = 5$ even | -0.506 | 0.795 | 40.8 | 44.3 | 47.1 |
| $10 = 5$ quantile | -2.232 | 0.442 | 33.5 | 33.9 | 33.7 |
| $10 = 5$ even | -0.537 | 0.795 | 40.4 | 44.4 | 47.6 |
| $20 = 5$ quantile | -2.214 | 0.444 | 33.5 | 33.6 | 33.6 |
| $30 = 5$ even | -0.500 | 0.802 | 40.6 | 44.5 | 47.1 |

## A.5 EFFECT OF CALIBRATION DATASET

In many cases, in-domain datasets are not available for calibration. To test robustness under this setting, we also perform calibration on a general instruction dataset, `alpaca-gpt4-en`. As shown in Table 5, performance with alpaca calibration is close to that obtained with in-domain data. While one might expect in-domain calibration to provide stronger correctness signals, alpaca still offers sufficiently reliable guidance.

A plausible reason out-of-domain calibration works is that the *mapping* from model confidence to expected correctness appears relatively stable across settings. In Figure 3, models of different sizes exhibit similar curves of expected accuracy as a function of confidence. Larger models place more mass in higher-confidence bins, which aligns with better benchmark scores. If this confidence–correctness relationship is driven by general properties of autoregressive modeling (rank-wise correctness decaying with rank) rather than dataset specifics, then a calibration set that matches the *format* of the target task (instruction→answer) may suffice even when its domain differs. We stress this is a hypothesis rather than a causal claim. We lack direct evidence that models possess an "intrinsic sense" of token-level correctness. Still, the observation that confidence–correctness curves are similar across model sizes suggests that correctness-aware calibration can transfer because it leverages structural, model-internal uncertainty signals rather than domain-specific features.

**Table 5:** Calibrated-TopK with $c_{CT} = 0.1$ using alpaca or GSM8K-train for calibration. Similar performances are observed. The test dataset is GSM8K-test.

| Model | alpaca-gpt4-en | | | GSM8K-train Set | | |
|---|---|---|---|---|---|---|
| | maj@8 | maj@16 | maj@32 | maj@8 | maj@16 | maj@32 |
| Qwen2.5-0.5B-Instruct | 40.1 | 44.3 | 46.5 | 41.0 | 45.7 | 47.4 |
| Qwen2.5-1.5B-Instruct | 72.4 | 74.8 | 76.1 | 72.9 | 74.8 | 76.5 |
| Qwen2.5-3B-Instruct | 79.9 | 80.8 | 81.0 | 79.1 | 80.7 | 80.7 |
| Qwen2.5-14B-Instruct | 92.9 | 93.4 | 93.3 | 93.0 | 93.4 | 93.8 |
| Qwen2.5-32B-Instruct | 92.6 | 93.3 | 93.6 | 92.9 | 93.3 | 93.3 |
| Llama-3.2-1B | 4.4 | 4.4 | 5.1 | 4.8 | 5.3 | 6.0 |
| Llama-3.2-1B-Instruct | 40.5 | 42.7 | 44.6 | 39.4 | 43.0 | 43.9 |

**Table 6:** MBPP performance by Qwen2.5-0.5B-Instruct and Qwen2.5-1.5B-Instruct. Using out-of-domain calibration dataset alpaca-gpt4-en results in better pass@k performance than using in-domain data.

| Method | Qwen2.5-0.5B-Instruct | | | Qwen2.5-1.5B-Instruct | | |
|---|---|---|---|---|---|---|
| | pass@8 | pass@16 | pass@32 | pass@8 | pass@16 | pass@32 |
| No restrictions | 41.6 | 50.1 | 57.6 | 55.6 | 65.2 | 72.5 |
| Top-k | 45.3 | 53.5 | 59.8 | 59.0 | 68.0 | 74.3 |
| Top-p | 47.0 | 55.3 | 63.0 | 60.2 | 68.6 | 74.9 |
| Min-p | **51.1** | 58.9 | 65.0 | 62.2 | 69.7 | 76.0 |
| EDT | 50.5 | 58.1 | 65.0 | 50.4 | 57.9 | 64.3 |
| $\eta$-sampling | 44.3 | 53.4 | 61.2 | 57.9 | 63.2 | 74.0 |
| Greedy-Threshold | 44.0 | 52.8 | 60.6 | 56.8 | 65.9 | 72.7 |
| $\varepsilon$-sampling | 49.2 | 56.1 | 61.8 | 62.0 | 69.6 | 75.4 |
| Calibrated-TopK (ID) | 50.3 | 57.9 | 64.2 | **62.6** | 69.3 | 74.6 |
| Calibrated-$\varepsilon$ (ID) | 48.6 | 53.8 | 57.8 | 62.1 | 68.2 | 72.8 |
| Calibrated-TopK (OOD) | 49.7 | 61.2 | **65.2** | 62.4 | 70.1 | 76.1 |
| Calibrated-$\varepsilon$ (OOD) | 51.0 | **61.4** | 64.2 | 62.3 | **70.4** | **76.3** |

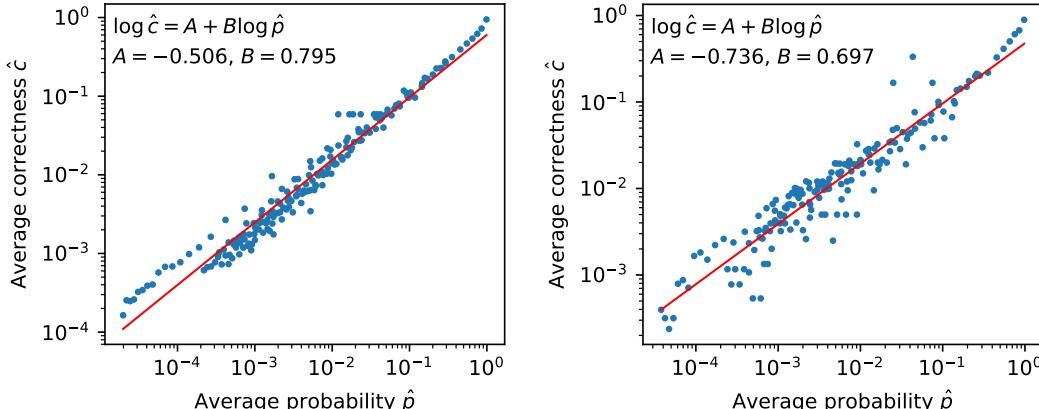

**(a)** GSM8K-train set calibration MSE loss= 0.134.    **(b)** MBPP-validation set calibration MSE loss= 0.237

**Figure 9:** Qwen2.5-05B-Instruct calibrated on MBPP-validation set has poorer linear fit than GSM8K, which might lead to worse performance.

## A.6 FAILURE CASE STUDY - POOR CALIBRATION SIGNALS

We provide a case study on MBPP (Austin et al., 2021), a code generation benchmark where our truncation strategies underperform compared to min-$p$ that prioritize diversity. Unlike reasoning tasks, MBPP requires directly producing executable code without intermediate steps, so every token is critical. A single incorrect token causes the program to fail its tests. Moreover, the pass@k metric rewards diversity, since success only requires one valid solution among the $k$ samples. In this setting, broader exploration increases the chance of producing at least one correct variant. Nevertheless, Greedy-Threshold still improves over unrestricted sampling, reinforcing our argument that sampling from low-confidence bins is harmful.

Calibration on MBPP also presents challenges. The validation set produces noisier signals, with a larger linear fit loss (Figure 9), which likely reduces the reliability of predicted rank-wise correctness and helps explain the weaker performance of Calibrated-$\varepsilon$ on Qwen2.5B-0.5B-Instruct. Despite this, Calibrated-TopK performs comparably to existing methods in the literature, suggesting that correctness-aware truncation remains useful even in diversity-driven domains. Additionally, if we use the less-noisy alpaca calibration variant, performance becomes better than using in-domain

calibration dataset and methods in literature too. This could be explained by in-domain dataset being 'overfit' that impose too strict sampling conditions as seen by higher gradient (B value). This results in more greedy-like behavior and better performance at pass@8 but worse behavior at pass@32 where diversity is more valuable.

To examine how the quality of the linear fit in calibration grids (log–log space) impacts performance, we plot the regression Mean Squared Error (MSE) alongside model accuracy in Figure 10. To isolate the effect of calibration quality from majority voting, we use maj@1 (single-sample accuracy). If the linear interpolation provides reliable correctness estimates, models should generate more accurate single completions. If the fit is poor, the predicted correctness signals may be uninformative or even harmful. To quantify this effect, we measure the improvement of Calibrated-$\varepsilon$ over the unrestricted baseline in maj@1. Figure 10 shows that as model size increases, regression MSE rises and the performance gain diminishes. While part of this effect reflects the general difficulty of improving already-strong large models, the trend also suggests that noisier calibration weakens the benefit of Calibrated-$\varepsilon$. This pattern is consistent with our previous observation, where MBPP's noisy validation signals lead Calibrated-$\varepsilon$ to underperform Calibrated-TopK. Importantly, however, maj@1 accuracy never drops below the no-sampling baseline, indicating that even poor calibration is not harmful.

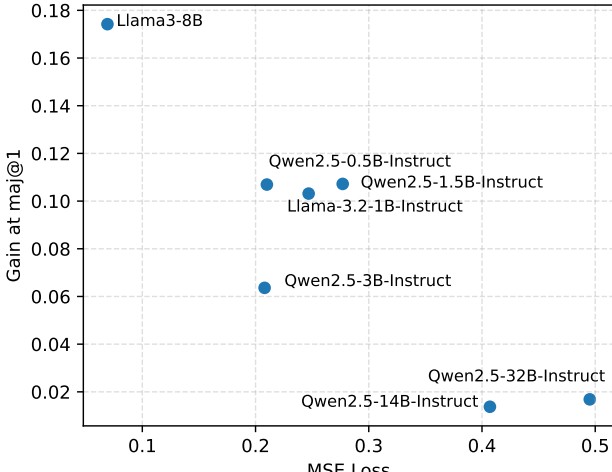

**Figure 10:** Effect of calibration fit quality. Larger models yield noisier linear fits (higher MSE), which correlates with smaller gains of Calibrated-$\varepsilon$ over the unrestricted baseline.

**Table 7:** Majority voting performance for Qwen2.5-1.5B-Instruct. Calibrated-TopK has the strongest overall performance.

| Method | GSM8K | | | MMLU-Pro | | | Big-Bench-Hard | | |
|---|---|---|---|---|---|---|---|---|---|
| | maj@8 | maj@16 | maj@32 | maj@8 | maj@16 | maj@32 | maj@8 | maj@16 | maj@32 |
| No restrictions | 68.4 | 71.5 | 73.1 | 32.9 | 34.4 | 35.4 | 38.9 | 41.8 | 43.0 |
| Top-k | 68.5 | 72.6 | 73.9 | 33.7 | 35.0 | 35.9 | 41.7 | 44.2 | 45.6 |
| Top-p | 71.1 | 74.1 | 75.9 | 34.1 | 35.4 | 36.4 | 44.0 | 46.2 | 47.1 |
| Min-p | 73.3 | 75.6 | 76.6 | 35.3 | 36.2 | 36.9 | 45.5 | 47.2 | 47.6 |
| EDT | **74.9** | 75.6 | 77.0 | 34.7 | 36.2 | 36.8 | 45.5 | **47.8** | 48.2 |
| $\eta$-sampling | 69.0 | 72.4 | 74.2 | 33.6 | 34.8 | 35.7 | 41.1 | 43.6 | 45.3 |
| $\varepsilon$-sampling | 73.4 | 76.4 | 78.2 | 35.0 | 36.2 | 36.9 | 45.4 | 46.9 | 47.7 |
| **Greedy-Threshold** | 70.1 | 73.6 | 75.5 | 33.6 | 34.9 | 36.1 | 40.3 | 42.9 | 44.4 |
| **Calibrated-TopK** | 72.3 | 75.2 | 76.3 | **35.6** | **36.4** | **37.0** | 45.8 | 47.7 | **48.5** |
| **Calibrated-$\varepsilon$** | 74.3 | **77.2** | **78.4** | 34.8 | 35.9 | 36.9 | 46.2 | 47.2 | 48.2 |

**Table 8:** Majority voted results on GSM8K and Big-Bench-Hard using Qwen2.5-1.5B-Instruct. In addition to existing sampling conditions, Greedy-Threshold $p_{GT} = 0.3$ is applied and shows strong consistent gains in addition to base samplers. Statistically significant differences ($p < 0.05$) marked in **bold**.

| Method | GSM8K | | | | Big-Bench-Hard | | | |
|---|---|---|---|---|---|---|---|---|
| | maj@1 | maj@8 | maj@16 | maj@32 | maj@1 | maj@8 | maj@16 | maj@32 |
| Baseline $T=1$ | 17.3 | 30.2 | 35.2 | 38.6 | 17.4 | 17.9 | 20.0 | 16.2 |
| + Greedy-Threshold | +0.3 | +1.0 | +1.8 | +2.0 | +1.9 | **+2.9** | **+0.3** | **+3.2** |
| top-$k$ | 18.8 | 32.6 | 38.7 | 41.9 | 20.5 | 22.0 | 21.7 | 21.5 |
| + Greedy-Threshold | +0.2 | **+1.1** | +0.5 | +1.1 | +1.1 | **+1.6** | **+1.5** | **+1.5** |
| top-$p$ | 22.4 | 35.5 | 40.8 | 43.6 | 22.3 | 25.5 | 25.8 | 25.9 |
| + Greedy-Threshold | +0.6 | +0.9 | +0.4 | +1.3 | +1.6 | **+1.5** | **+1.5** | **+1.9** |
| min-$p$ | 25.3 | 38.7 | 43.1 | 46.6 | 27.5 | 30.6 | 31.5 | 31.7 |
| + Greedy-Threshold | +1.9 | +1.4 | +0.8 | +0.6 | +0.3 | +0.2 | +0.1 | +0.1 |
| EDT | 28.0 | 40.2 | 44.7 | 46.8 | 27.0 | 30.4 | 31.1 | 31.7 |
| + Greedy-Threshold | 0.0 | +0.2 | -0.3 | +0.1 | +0.5 | +0.3 | +0.5 | **+0.3** |
| $\eta$-sampling | 19.0 | 31.6 | 37.2 | 41.0 | 19.7 | 20.6 | 20.3 | 19.6 |
| + Greedy-Threshold | +4.3 | **+2.4** | **+1.8** | **+1.7** | **+0.9** | **+1.9** | **+2.2** | **+2.7** |

## A.7 FURTHER CALIBRATED TRUNCATION AND GREEDY-THRESHOLD RESULTS

We further compare our proposed methods against other existing methods across different benchmarks. Calibrated-TopK has the strongest overall performance for Qwen2.5-1.5B-Instruct as shown in Table 8.

We extend our analysis of Greedy-Threshold for up to 32B parameters models in Table 9, considering both instruct and non-instruct models. As expected, larger models with stronger baseline performance are more challenging to improve. Existing samplers provides little improvement on the baseline. Nevertheless, Greedy-Threshold does not degrade performance. It either provides modest gains or remains comparable to existing samplers. One explanation for this diminishing effect is that larger models produce high-confidence predictions more frequently (Figure 3), leading to fewer low-confidence steps. Since Greedy-Threshold only intervenes under low-confidence conditions, its impact naturally diminishes as model size increases. Similarly, the effect of majority voting in larger models (Table 11) also diminishes in larger models due to reduced stochasticity.

**Table 9:** Majority voted results on GSM8K. Greedy-Threshold improves performance more in smaller models. Statistically significant difference ($p < 0.05$) marked in **bold**.

| Method | Llama-3.2-1B | | | Qwen2.5-14B-Instruct | | | Qwen2.5-32B-Instruct | | |
|---|---|---|---|---|---|---|---|---|---|
| | maj@8 | maj@16 | maj@32 | maj@8 | maj@16 | maj@32 | maj@8 | maj@16 | maj@32 |
| Baseline $T$=1 | 0.7 | 0.6 | 0.2 | 92.8 | 93.4 | 93.5 | 92.8 | 93.4 | 94.0 |
| + Greedy-Threshold | +0.7 | **+0.5** | +0.5 | +0.2 | +0.2 | **+0.3** | +0.1 | +0.3 | -0.2 |
| Top-$k$ | 2.6 | 1.7 | 1.4 | 93.3 | 93.5 | 93.7 | 92.9 | 93.7 | 93.9 |
| + Greedy-Threshold | -0.3 | +0.6 | +0.1 | 0.0 | -0.1 | 0.0 | +0.1 | -0.1 | 0.0 |
| Top-$p$ | 1.7 | 1.4 | 1.6 | 93.3 | 93.5 | 93.5 | 93.1 | 93.5 | 93.7 |
| + Greedy-Threshold | +0.3 | +0.5 | +0.4 | +0.1 | -0.1 | 0.0 | -0.2 | -0.1 | -0.2 |
| Min-$p$ | 3.9 | 3.9 | 3.6 | 93.2 | 93.1 | 93.3 | 92.8 | 93.4 | 93.6 |
| + Greedy-Threshold | +0.9 | +0.5 | +0.9 | -0.3 | +0.2 | +0.1 | +0.3 | 0.0 | -0.2 |
| EDT | 4.2 | 3.8 | 3.9 | 92.9 | 93.3 | 93.4 | 92.7 | 93.3 | 93.5 |
| + Greedy-Threshold | +0.9 | +0.3 | +0.1 | +0.1 | -0.1 | 0.0 | -0.1 | -0.1 | +0.1 |
| $\eta$-sampling | 1.1 | 0.8 | 0.5 | 93.5 | 93.6 | 93.7 | 93.2 | 93.6 | 93.9 |
| + Greedy-Threshold | +0.7 | +0.5 | +0.7 | -0.4 | -0.3 | +0.1 | -0.1 | 0.0 | -0.1 |

**Table 10:** GSM8K performance by Qwen2.5-7B and Qwen2.5-7B-Instruct.

| Method | Qwen2.5-7B | | | Qwen2.5-7B-Instruct | | |
|---|---|---|---|---|---|---|
| | pass@8 | pass@16 | pass@32 | pass@8 | pass@16 | pass@32 |
| No restrictions | 82.8 | 86.8 | 88.4 | 86.7 | 88.4 | 89.4 |
| Top-k | 83.3 | 87.6 | 88.9 | 87.2 | 89.3 | 89.3 |
| Top-p | 84.9 | 88.2 | 89.6 | 87.2 | 88.3 | 89.3 |
| Min-p | 86.2 | **88.7** | **89.9** | 87.2 | 88.3 | 88.8 |
| EDT | 83.4 | 87.3 | 88.7 | 87.0 | 89.1 | **89.6** |
| $\eta$-sampling | 83.8 | 87.2 | 89.5 | 87.8 | 89.3 | 89.7 |
| Greedy-Threshold | 83.8 | 86.6 | **88.9** | 86.8 | 88.8 | **89.6** |
| $\varepsilon$-sampling | **86.6** | 88.5 | **89.9** | 87.4 | 88.7 | 88.8 |
| Calibrated-TopK | 86.5 | **88.7** | **89.9** | 87.4 | **89.6** | **89.6** |
| Calibrated-$\varepsilon$ | 86.5 | 87.6 | 89.3 | **88.0** | 88.7 | 89.4 |

**Table 11:** MMLU-Pro performance by Qwen2.5-14B-Instruct and Qwen2.5-32B-Instruct.

| Method | Qwen2.5-14B-Instruct | | | Qwen2.5-32B-Instruct | | |
|---|---|---|---|---|---|---|
| | pass@8 | pass@16 | pass@32 | pass@8 | pass@16 | pass@32 |
| No restrictions | 63.4 | 63.7 | 64.1 | 68.1 | 68.5 | 68.7 |
| Greedy-Threshold | 63.4 | 64.1 | 64.0 | 68.1 | 68.5 | 68.8 |
| $\varepsilon$-sampling | 63.6 | **64.2** | 64.2 | **68.6** | 68.9 | 68.9 |
| Calibrated-TopK | **63.7** | **64.2** | **64.6** | 68.3 | 68.7 | **69.0** |
| Calibrated-$\varepsilon$ | 63.8 | 63.7 | 64.1 | 68.3 | 68.6 | 68.9 |

A.8   EFFECTIVENESS AT LOW TEMPERATURES

Since lower temperatures are used for math and coding tasks, we test our proposed methods using $T = 0.6$. The reason for temperature scaling is to make probability distribution more peaked so that top-1 probability increases while the tail probabilities shrink. Consequently, probability-based pruning becomes implicitly more aggressive. Many low-rank tokens become unlikely to be sampled even without changing any thresholds. This aligns with our thesis that low ranked tokens should not be broadly sampled due to their correlation with low correctness. Given the same non top-1 ranked token, after temperature scaling, its probability will decrease. Thus, to remove the same token as without temperature scaling, the probability cutoff needs to be lower. We choose $\varepsilon = 0.01$, $p_{GT} = 0.1$ and $c_{CT} = 0.01$.

Given the same $p_{max}$, its scaled probability would be bigger. Thus, we should set a higher Greedy-Threshold than before. However, in practice we found that this limits diversity significantly that the benefit from self-consistency diminishes. The maj@1 accuracy increases but performance gain from maj@k reduces. We hypothesize that this is because more lower temperature already results in diminished diversity. Further restrictions results in diversity collapse. From Table 12, we can see that further gains from baseline is much smaller than with $T = 1$. Nevertheless, by setting lower truncation thresholds, we still see performance gains from using our proposed truncation methods.

**Relation between temperature and $\varepsilon$-sampling.**   We derive how temperature scaling interacts with probability thresholding in $\varepsilon$-sampling. At a fixed decoding step $t$, let $z_t(j)$ denote the logit of token $j$ and $p_T(j)$ the corresponding next-token probability under temperature $T > 0$,

$$p_T(j) \; = \; \frac{\exp\big(z_t(j)/T\big)}{\sum_v \exp\big(z_t(v)/T\big)}. \tag{12}$$

Let $j^\star$ be the top-1 token at this step and define centered logits $\Delta z_j := z_t(j) - z_t(j^\star) \leq 0$. Subtracting $z_t(j^\star)$ from all logits leaves the softmax invariant, so

$$p_T(j) \; = \; \frac{\exp(\Delta z_j/T)}{1 + \sum_{k \neq j^\star} \exp(\Delta z_k/T)} \; = \; \frac{\exp(\Delta z_j/T)}{D_T}, \tag{13}$$

where we introduced the normalizer

$$D_T := 1 + \sum_{k \neq j^\star} \exp(\Delta z_k/T). \tag{14}$$

In particular, for $T = 1$ we have

$$p_1(j) \; = \; \frac{\exp(\Delta z_j)}{D_1}, \qquad D_1 := 1 + \sum_{k \neq j^\star} \exp(\Delta z_k). \tag{15}$$

We can now eliminate the logits $\Delta z_j$ and obtain a direct relation between $p_T(j)$ and $p_1(j)$ at the same decoding step. From the $T = 1$ expression we get

$$\exp(\Delta z_j) = p_1(j)\, D_1, \quad \Rightarrow \quad \exp(\Delta z_j/T) = \big(\exp(\Delta z_j)\big)^{1/T} = \big(p_1(j)\, D_1\big)^{1/T}. \tag{16}$$

**Table 12:** Majority voted results on GSM8K with scaled temperature $T = 0.6$, $\varepsilon = 0.01$, $p_{GT} = 0.1$ and $c_{CT} = 0.01$

| Method | Qwen2.5-0.5B-Instruct | | | Qwen2.5-1.5B-Instruct | | |
|---|---|---|---|---|---|---|
| | maj@8 | maj@16 | maj@32 | maj@8 | maj@16 | maj@32 |
| No conditions | 40.7 | 44.9 | 46.9 | 73.7 | 76.3 | 76.9 |
| $\varepsilon$-sampling | **41.4** | **45.2** | 46.8 | **74.4** | 75.8 | 77.0 |
| **Greedy-Threshold** | 41.2 | **45.2** | 47.2 | 74.3 | **76.9** | 77.1 |
| **Calibrated-TopK** | 40.9 | 44.5 | 46.7 | 74.1 | 76.0 | **77.3** |
| **Calibrated-$\varepsilon$** | **41.4** | **45.2** | 48.4 | 74.2 | 75.6 | 77.0 |

Plugging this into the definition of $p_T(j)$ gives

$$p_T(j) \;=\; \frac{\left(p_1(j)\,D_1\right)^{1/T}}{D_T} \;=\; \underbrace{\frac{D_1^{1/T}}{D_T}}_{K_T}\, p_1(j)^{1/T}. \tag{17}$$

Thus, for any fixed decoding step $t$ and any temperature $T > 0$,

$$p_T(j) = K_T\, p_1(j)^{1/T}, \tag{18}$$

where the factor

$$K_T := \frac{D_1^{1/T}}{D_T} \tag{19}$$

depends on the full logit configuration at that step and on $T$, but does not depend on the particular token $j$. In other words, temperature rescales per-token probabilities via a power law in their $T = 1$ probabilities, up to a step-wise constant multiplier $K_T$ shared by all tokens. This leads to a natural scaling rule for $\varepsilon$-sampling. Suppose that at $T = 1$ we apply $\varepsilon_1$-sampling and discard all tokens with

$$p_1(j) \;\leq\; \varepsilon_1. \tag{20}$$

The same tokens have probability, at temperature $T$,

$$p_T(j) \;=\; K_T\, p_1(j)^{1/T} \;\leq\; K_T\, \varepsilon_1^{1/T}. \tag{21}$$

If we want our temperature-$T$ cutoff to remove at least all the tokens that would have been removed by $\varepsilon_1$-sampling at $T = 1$, a natural choice is

$$\varepsilon_T \propto \varepsilon_1^{1/T}, \tag{22}$$

with the proportionality constant absorbing an average over the step-wise factors $K_T$. In practice we use the simple global scaling rule

$$\varepsilon_T \approx \varepsilon_1^{1/T}. \tag{23}$$

For $T < 1$ (sharper distributions), $1/T > 1$ and hence $\varepsilon_T < \varepsilon_1$. Temperature already suppresses low-probability tokens, so the probability threshold must be lowered in order to prune a comparable part of the tail of the logit distribution.

**Intuition for the $\varepsilon = 0.01$ choice.** In our main experiments we use $\varepsilon_1 = 0.05$ at $T = 1$. Applying Equation (23) to $T = 0.6$ yields the theoretical value

$$\varepsilon_{0.6}^{\text{theory}} \;\approx\; \varepsilon_1^{1/0.6} = 0.05^{1/0.6} \approx 6.8 \times 10^{-3}. \tag{24}$$

For our low-temperature runs at $T = 0.6$ we instead adopt $\varepsilon = 0.01$, which is slightly more conservative (it removes a bit more of the tail) but remains close to the theoretical prediction.

**Table 13:** GPT-4-mini judges performance of various samplers on LitBench prompts generated by Qwen2.5-7B-Instruct. Human written baseline is the chosen_story column from the original benchmark. Each category is scored from 1-5. Higher is better.

| Method | Relevance | Coherence | Emotional impact | Originality | Total |
|---|---|---|---|---|---|
| Greedy | 4.50 | 3.27 | 2.70 | 2.68 | 13.15 |
| $T{=}1$ | 4.48 | 3.15 | 2.81 | 2.67 | 13.11 |
| Min-$p$=0.1 | 4.48 | 3.38 | 2.80 | 2.69 | 13.35 |
| Human written baseline | 4.50 | 3.81 | 3.03 | 3.20 | 14.54 |
| Calibrated-TopK | 4.58 | 3.52 | 2.77 | 2.68 | 13.55 |
| Calibrated-$\varepsilon$ | 4.60 | 3.47 | 2.77 | 2.69 | 13.53 |

## A.9    WHAT ABOUT CREATIVE WRITING?

Our main experiments target reasoning tasks with closed-form answers. A natural question is whether the same correctness-aware perspective applies to open-ended generation. We provide an illustrative case study on creative writing using LitBench (Fein et al., 2025), treating the *chosen* story as a proxy for ground truth. We construct calibration grids and probability–correctness scatter plots and observe qualitatively similar patterns. As confidence decreases, expected correctness declines. As rank increases, correctness drops sharply. Compared with GSM8K and Alpaca calibration, creative prompts exhibit systematically lower confidence, with low-confidence bins occurring more often (e.g., lowest-bin frequency: $6.01\,\%$ for LitBench vs. $0.17\,\%$ for Alpaca and $0\,\%$ for GSM8K). The probability–correctness mapping in log–log space is also stronger in this setting, with a slope closer to one. Full calibration diagrams and scatter plots are provided in Section A.10.

We conduct a simple evaluation of our samplers compared to min-p (Schaeffer et al., 2025), which is advertised to excel at creative writing. We prompt Qwen2.5-7B-Instruct to generate short stories using the prompts given in LitBench with the following prompt: *Given the following writing prompt, write a short story that is original, relevant, emotional and coherent correct in less than 500 words. [Prompt] Your story:*

The stories generated are evaluated by GPT-5-mini [9] using the following prompt: *Evaluate a creative writing task and give scores. Each category is scored from 1 (lowest) to 5 (highest). Consider these categories: Originality: unique concepts, unexpected elements. Relevance: story follows the writing prompt. Emotional impact: how the writing affects the reader. Coherence: logical flow and narrative structure. Writing prompt: [prompt] Judge the writing in the following format: Reasoning: [your evaluation with scores for each category]*

The scores given by the judge model is parsed and averaged for each category averaged over three runs. As shown in Table 13, our calibrated samplers outperforms min-p and no-samplers baseline by improving coherence and relevance. While originality stays the same and emotional impact is slightly reduced. The improvement in coherence and relevance is essential for a small model like Qwen2.5-7B-Instruct, which is prone to drifting off-topic at a high temperature.

## A.10    EXAMPLE CALIBRATION DIAGRAMS

---

[9] https://openrouter.ai/openai/gpt-5-mini

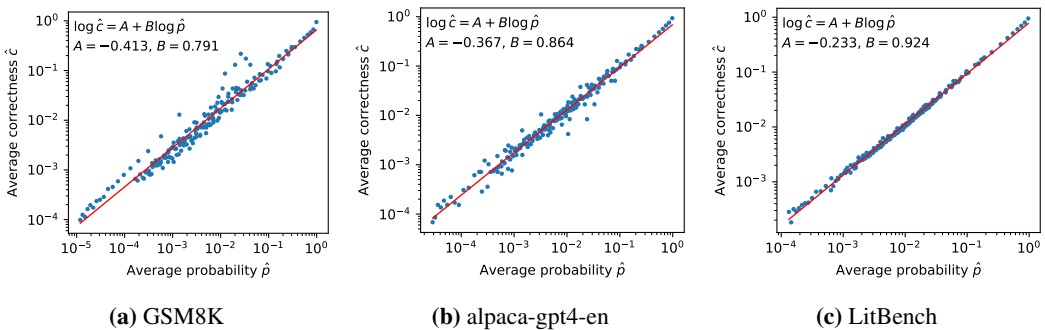

**(a)** GSM8K  **(b)** alpaca-gpt4-en  **(c)** LitBench

**Figure 11:** Calibration scatter plots on Qwen2.5-1.5B-Instruct

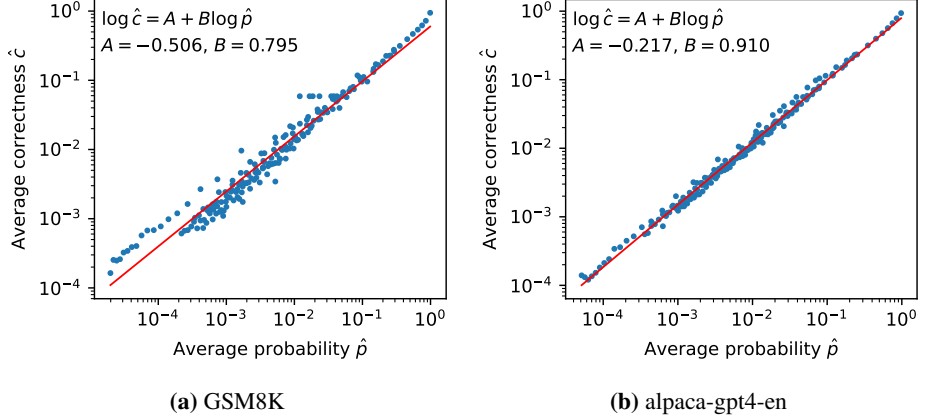

**(a)** GSM8K  **(b)** alpaca-gpt4-en

**Figure 12:** Calibration scatter plots on Qwen2.5-0.5B-Instruct

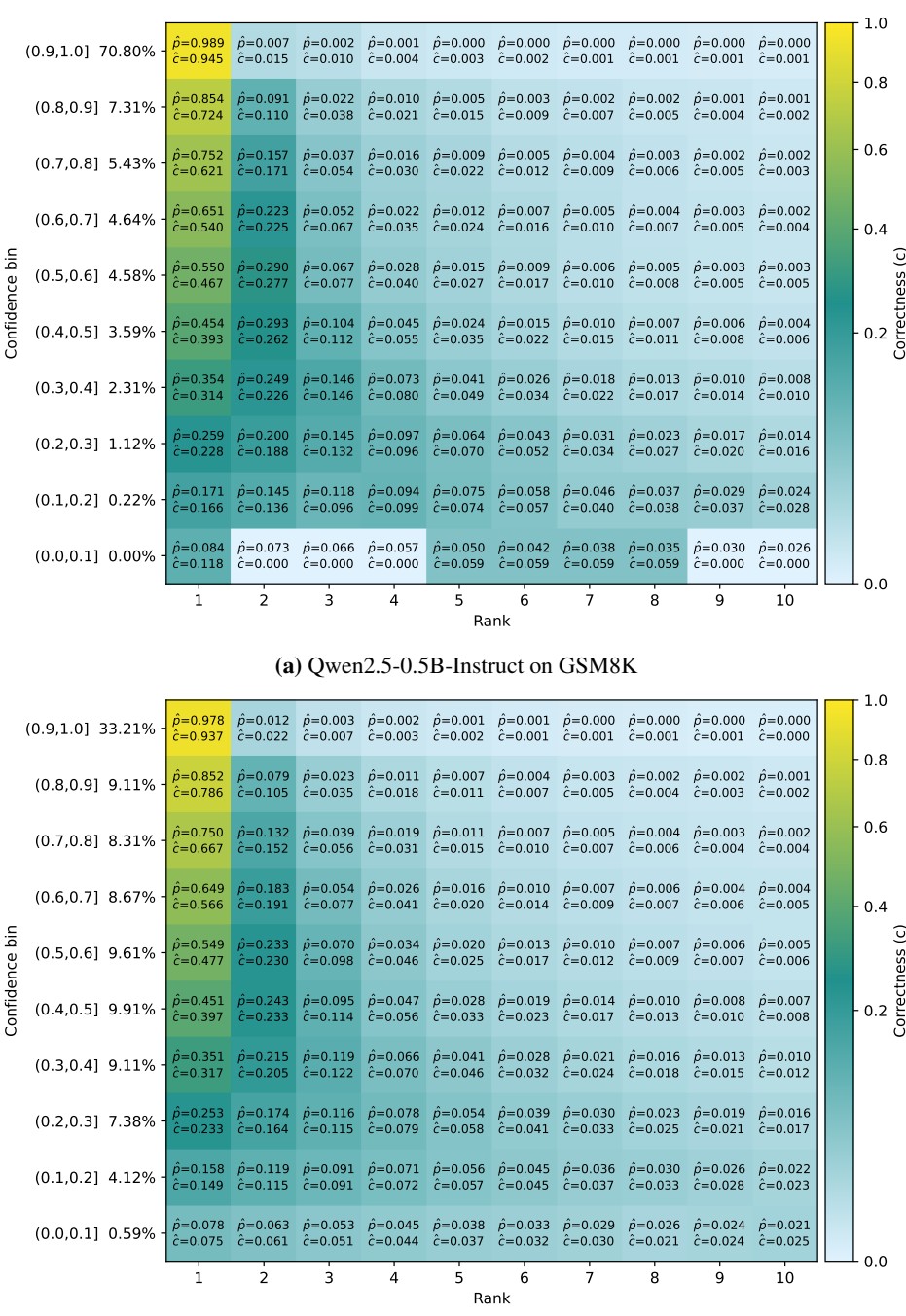

**(a)** Qwen2.5-0.5B-Instruct on GSM8K

**(b)** Qwen2.5-0.5B-Instruct on alpaca-gpt4-en

**Figure 13:** Calibration grids on various Qwen models and GSM8K or alpaca-gpt4-en

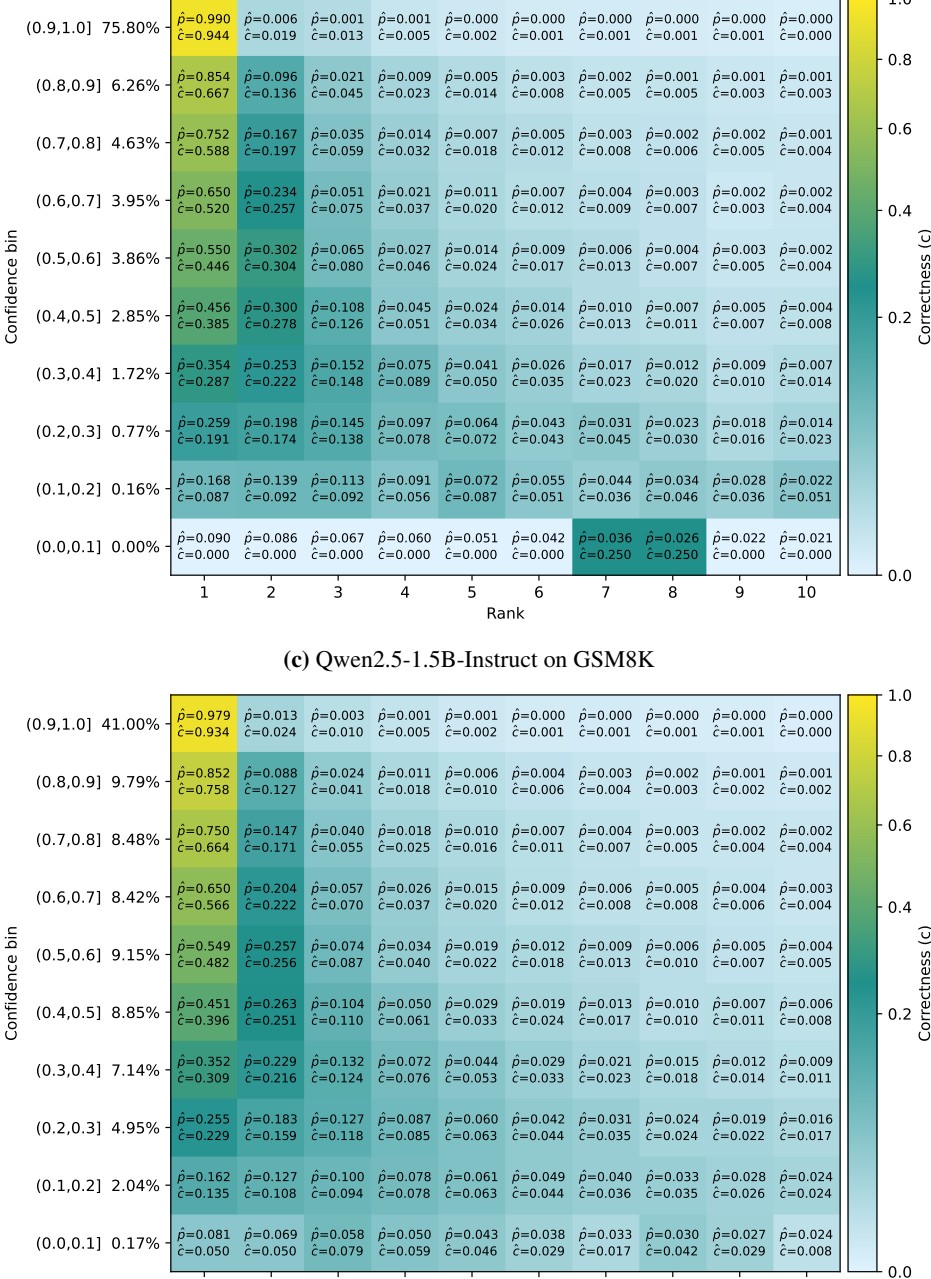

**(c)** Qwen2.5-1.5B-Instruct on GSM8K

**(d)** Qwen2.5-1.5B-Instruct on alpaca-gpt4-en

**Figure 13:** Calibration grids on various Qwen models and GSM8K or alpaca-gpt4-en

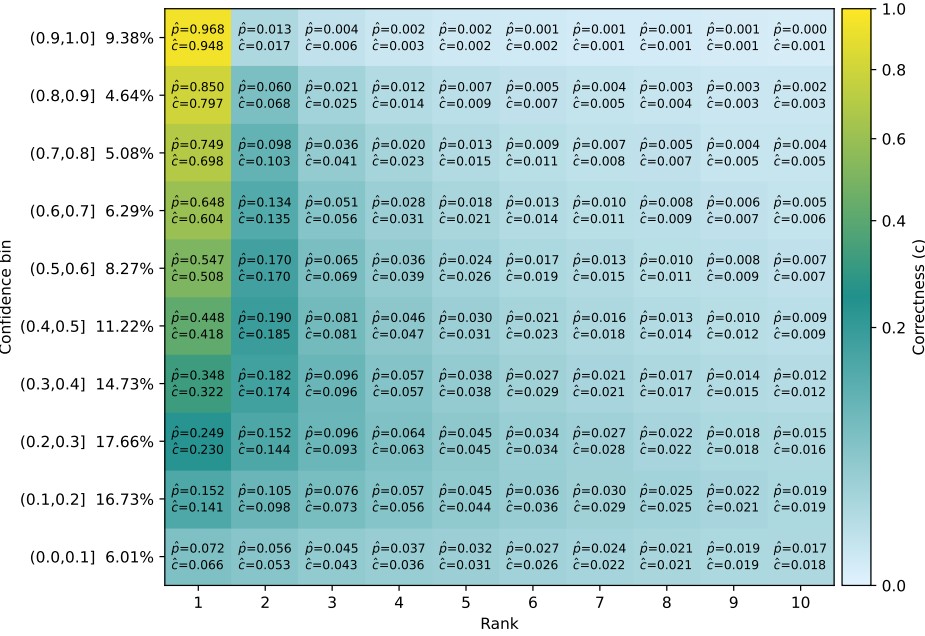

**(e)** Qwen2.5-1.5B-Instruct on LitBench

**Figure 13:** Calibration grids on various Qwen models and GSM8K, alpaca-gpt4-en or LitBench

