# OpenReview forum: "Sample Smart, Not Hard: Correctness-First Decoding for Better Reasoning in LLMs"
_ICLR.cc/2026/Conference — ICLR 2026 Poster_

### Official Review · Reviewer_cpeR · 2025-10-30

**Soundness:** 3
**Presentation:** 3
**Contribution:** 3
**Rating:** 4
**Confidence:** 2

**Summary:**

The paper proposes a new method that samples based on correctness during the decoding process. The authors conduct comprehensive experiments to demonstrate the effectiveness of their approach.

**Strengths:**

1. The idea is both novel and elegant. The authors approach the problem from a fresh perspective and achieve the goal of evaluating correctness without relying on complex architectures.

2. The motivation behind the work is strong, and the authors provide a comprehensive and convincing analysis.

3. The paper is well written and easy to follow.

**Weaknesses:**

1. The experimental results indicate only marginal improvements. In the main results table, the accuracy gain is approximately 1%, which appears to be relatively minor.

2. It's valuable that the paper includes results on GPT-OSS; however, the evaluations on Qwen are limited to smaller model sizes. It would strengthen the paper to include results on larger models, such as the 7B or 8B variants, or even larger if available.

**Questions:**

NA

---

> ### Author Response · Authors · 2025-11-17
> **Additional larger model experiments**
>
> Thank you for your feedback\! We’re glad you find the motivation and analysis of our work convincing and find that it “achieve the goal of evaluating correctness without relying on complex architectures”.
>
> * **Marginal performance gains.** We agree that some of the gains may look modest. However, our method is learning-free with zero additional inference cost. We do not update weights, architecture or increase forward pass computations. This makes large jumps in accuracy unlikely but also means consistent gains are essentially “free” at inference time. Additionally, we note that in Table 1, we do see that performance increases noticeably on GSM8k, at about 10%, over standard sampling. The gap to other samplers is smaller, but this is in line with our framework. Other samplers, which end up heuristically implementing a similar strategy (see the strategy comparison in Fig.1), also work well. Our work, aside from modestly improving these samplers, also provides a principled way to understand why these sampling strategies improve performance. In Table 2, we note that we only show that the simplest way of combining our findings with existing samplers, namely to implement a fixed greedy threshold, already improves performance. Since the cost of our simple sampling method is negligible, any consistent improvement is valuable.
> * **Small model sizes.** We appreciate the suggestion to evaluate larger models. Our current submission already includes experiments on Qwen2.5-14B-Instruct and Qwen2.5-32B-Instruct in Appendix A.7. We now additionally provide results on Qwen2.5-7B and Qwen2.5-7B-Instruct on GSM8K, and 14/32B performance on MMLU-Pro below. These are also added to Appendix A.7. For these comparisons, we want to highlight that our truncation thresholds behave differently mainly when model systemic confidence is low (Figure 1\) compared to methods in literature. Thus, larger improvements are observed for smaller models with generally lower confidence (Figure 3\) and a higher chance for epistemic error. It is entirely expected in our framework that minor improvements are observed for larger models, which are generally more confident, and are at a lower risk of making epistemic mistakes.
> * Table: GSM8K performance by Qwen2.5-7B and Qwen2.5-7B-Instruct.
>
> |  |  | 7B |  |  | 7B-Instruct |  |
> | :---- | :---: | :---: | :---: | :---: | :---: | :---: |
> |  | maj@8 | maj@16 | maj@32 | maj@8 | pass@16 | pass@32 |
> | Baseline | 0.828 | 0.868 | 0.884 | 0.867 | 0.884 | 0.894 |
> | Top-k | 0.833 | 0.876 | 0.889 | 0.872 | 0.893 | 0.893 |
> | Top-p | 0.849 | 0.882 | 0.896 | 0.872 | 0.883 | 0.893 |
> | Min-p | 0.862 | **0.887** | **0.899** | 0.872 | 0.883 | 0.888 |
> | EDT | 0.834 | 0.873 | 0.887 | 0.870 | 0.891 | **0.896** |
> | μ-sampling | 0.838 | 0.872 | 0.895 | 0.878 | 0.893 | 0.897 |
> | Greedy-Threshold | 0.838 | 0.866 | **0.889** | 0.868 | 0.888 | **0.896** |
> | ε-sampling | **0.866** | 0.885 | **0.899** | 0.874 | 0.887 | 0.888 |
> | Calibrated-TopK  | 0.865 | **0.887** | **0.899** | 0.874 | **0.896** | **0.896** |
> | Calibrated-ε | 0.865 | 0.876 | 0.893 | **0.880** | 0.887 | 0.894 |
>
> * Table: MMLU-Pro performance by Qwen2.5-14B-Instruct and Qwen2.5-32B-Instruct.
>
> |  |  | 14B |  |  | 32B |  |
> | :---- | :---: | :---: | :---: | :---: | :---: | :---: |
> |  | maj@8 | maj@16 | maj@32 | maj@8 | pass@16 | pass@32 |
> | Baseline | 63.4 | 63.7 | 64.1 | 68.1 | 68.5 | 68.7 |
> | Greedy-Threshold | 63.4 | 64.1 | 64.0 | 68.1 | 68.5 | 68.8 |
> | ε-sampling | 63.6 | **64.2** | 64.2 | **68.6** | 68.9 | 68.9 |
> | Calibrated-TopK  | **63.7** | **64.2** | **64.6** | 68.3 | 68.7 | **69.0** |
> | Calibrated-ε | 63.8 | 63.7 | 64.1 | 68.3 | 68.6 | 68.9 |
>
> Overall, thank you for your comments. Please let us know if there are any follow-up questions or concerns. We’d be happy to address them.

---

> > ### Author Response · Authors · 2025-11-26
> >
> > Dear reviewer, we'd like to kindly follow up on our responses to your reviews. We would greatly appreciate it if you can let us know whether our responses address your concerns regarding model sizes and benchmark performances, or if any further clarification is needed.

---

### Official Review · Reviewer_2drS · 2025-10-31

**Soundness:** 2
**Presentation:** 2
**Contribution:** 2
**Rating:** 4
**Confidence:** 2

**Summary:**

paper presents an interesting and ambitious analysis of Correctness-First Decoding for Better Reasoning in LLMs. It’s well-structured and follows a logical argument, but several areas could be strengthened for clarity, rigor, and reader engagement.

**Strengths:**

1. The introduction effectively frames the research question and contextualizes the problem within existing literature.
2. The Methods section provides an appropriate research design and statistical treatment. The inclusion of comparative baselines is a strong point.
3. The experimental design and analytical approach are sound and appropriate for the research question.

**Weaknesses:**

1. The abstract summarizes methods more than findings; it doesn’t highlight the key quantitative results or main contributions.
2. Some information about sample selection, data splits, or parameter settings is missing, which could affect reproducibility.
3. The discussion section mostly restates results instead of analyzing their implications or addressing possible alternative explanations.
4. The conclusion doesn’t emphasize broader implications or concrete future directions, making the ending feel abrupt.

**Questions:**

N/A

---

> ### Author Response · Authors · 2025-11-17
> **Writing improvements**
>
> We thank the reviewer for their suggestion for writing and structure improvements in the paper. We have updated the PDF to improve the abstract, settings, discussion and conclusion.
>
> * **Abstract quantitative results.** We edited the abstract to highlight improved quantitative results on AIME by up to 6%.
> * **Lack of detailed experiment setup description**. Due to length restriction, all the detailed setups are described in Appendix A.1. However, we have added a description of data split and comparison parameter settings in the main body, to make our experimental setup self-contained.
> * **Lack of in depth discussion**. We have added a new, detailed, discussion of calibration noise, sparsity and cross-domain generalization with additional example figures in page 9. We discuss that cross-domain generalization is advantageous by calibrating on a general-purpose instruction dataset (alpaca-gpt4-en). However, if we use an in-domain dataset that does not provide enough samples, then the calibration signal might be poor and noisy, which reduces performance. This also relates to model overconfidence for certain tasks. If the evaluation metric encourages diversity, then overconfident calibration has disadvantages, and it is better to switch to out-of-domain calibration dataset.
> * **Conclusion brevity**. We have added a future directions paragraph in the conclusion section, and discuss broader implications more. We discuss future directions such as open-ended generation, online calibration and cross-model calibration comparisons.
>
> Overall, we’re glad to receive this type of feedback, and we think this has improved the writing of our submission. Please feel free to take a look at the updated PDF. If the changes are acceptable, and given that there seem to be no concerns regarding the technical content of our submission, we would appreciate it if you would consider updating your score.

---

> ### Author Response · Authors · 2025-11-26
>
> We are writing to follow up on our rebuttal and responses to your review. We would greatly appreciate it if you could indicate whether our paper writing and structural updates are sufficient, or if additional modifications would be helpful.

---

### Official Review · Reviewer_7oG6 · 2025-11-01

**Soundness:** 3
**Presentation:** 3
**Contribution:** 3
**Rating:** 6
**Confidence:** 4

**Summary:**

The paper argues that current decoding for reasoning mixes up two kinds of uncertainty:  good uncertainty (many valid continuations) and bad uncertainty (the model is just wrong). So instead of sampling more when the model is low-confidence, they propose “correctness-first” decoding: Greedy-Threshold (go fully greedy when max prob < τ), Calibrated-TopK (adapt k per step using a confidence×rank correctness grid), and Calibrated-ε (a continuous version that maps prob to expected correctness).On GSM8K, MMLU-Pro, BBH, and AIME (with GPT-OSS) these rules consistently give small-but-real maj@k / pass@k gains and can be layered on top of normal samplers which I found to be good.

**Strengths:**

- Very clear problem framing
- Easy to implement and add to existing samplers and inference is cheap
- The experiments were on broader side and showed consistency across range of models

**Weaknesses:**

- The paper seems to relies on task/data calibration; when calibration is noisy or OOD, gains shrink , so it’s not always free wins.
- From the paper it seems like improvements are incremental, few points  maj@k, big models already strong.
- The claim low confidence = epistemic, so always sample less is shown mainly on math/reasoning; creative/open-ended generation is only discussed, not tested

**Questions:**

Please refer to weaknesses

---

> ### Author Response · Authors · 2025-11-17
> **OOD generalization and creative writing**
>
> We thank the reviewer for the positive assessment of our clear problem framing and understanding of our cheap inference method.
>
> * **Noisy calibration and OOD concerns.** The data calibration is indeed a noticeable  influence especially with training sets that do not have enough data points. We added more detailed discussion about noise and cross-domain generalization in the main discussion section (page 9). Yet, we find that cross-domain generalization from general purpose instruction tuning datasets to maths/coding is still good, if no data is available for a specific domain. Thus, we suggest that if in-domain datasets are noisy or sparse, it is a good idea to revert to using a general-purpose dataset, like generic instruction data. **Here we provide an example that generic instruction data, which is out-of-distribution for this code task, can still result in good performance.** This is also added to Appendix A.5 and A.6.
> * Table: MBPP evaluated on Qwen2.5-0.5B-Instruct and Qwen2.5-1.5B-Instruct using in-distribution (MBPP-train) and out-of-distribution (alpaca-gpt4-en) calibration datasets.
>
> |          | |    0.5B    |  | |    1.5B    |        |
> | :---- | :---: | :---: | :---: | :---: | :---: | :---: |
> |  | maj@8 | maj@16 | maj@32 | maj@8 | pass@16 | pass@32 |
> | Baseline | 41.6 | 50.1 | 57.6 | 55.6 | 65.2 | 72.5 |
> | Top-k | 45.3 | 53.5 | 59.8 | 59 | 68 | 74.3 |
> | Top-p | 47.0 | 55.3 | 63.0 | 60.2 | 68.6 | 74.9 |
> | Min-p | **51.1** | 58.9 | 65.0 | 62.2 | 69.7 | 76 |
> | EDT | 50.5 | 58.1 | 65.0 | 50.4 | 57.9 | 64.3 |
> | μ-sampling | 44.3 | 53.4 | 61.2 | 57.9 | 63.2 | 74 |
> | Greedy-Threshold | 44.0 | 52.8 | 60.6 | 56.8 | 65.9 | 72.7 |
> | ε-sampling | 49.2 | 56.1 | 61.8 | 62 | 69.6 | 75.4 |
> | Calibrated TopK (ID) | 50.3 | 57.9 | 64.2 | **62.6** | 69.3 | 74.6 |
> | Calibrated-ε (ID) | 48.6 | 53.8 | 57.8 | 62.1 | 68.2 | 72.8 |
> | Calibrated-TopK (OOD) | 49.7 | 61.2 | **65.2** | 62.4 | 70.1 | 76.1 |
> | Calibrated-ε  (OOD) | 51 | **61.4** | 64.2 | 62.3 | **70.4** | **76.3** |
>
> * **Concerns of incremental improvements.** We agree that some of the gains may look incremental. However, our method is **learning-free with zero additional inference cost**. We do not update weights, architecture or increase forward pass computation. This makes large jumps in accuracy unlikely but also means consistent gains are essentially “free” at inference time. Additionally, large(r) models are more confident in general, meaning less stochasticity is available for majority voting. Further, for a fixed task complexity, larger models have less systemic uncertainty (they are more often correct), so modifying their sampler to reduce systematic uncertainty is less impactful. This is in line with our framework, which predicts that errors are caused by systemic uncertainty. Thus, it is natural to not see large gains from sampling, especially for very large models.
> * **Lack of creative writing testing**. The original focus of our paper was on reasoning tasks where even a single epistemic error could break the model’s response.  As such, we relegated the impact of different calibration behavior for creative writing to Appendix A.9. However, we’ve now extended this section and **added a quantitative experiment on creative writing using LitBench and LLM-as-a-judge evaluations**. We compare it to min-p, which is a sampler advertised to excel at creative writing. We see that with our samplers, story relevance and coherence are improved. Thus, interestingly, our samplers help the story generations to stay grounded and coherent, which is very important for small models like Qwen2.5-7B-Instruct which was not designed for creative writing. This is in-line with our claim that epistemic errors can be avoided by not sampling low-probability tokens. We’ve added details regarding this experiment to Appendix A.9.
> * Table: GPT-4-mini judging the performance of various samplers on LitBench prompts generated by Qwen2.5-7B-Instruct. The human written baseline is the chosen\_story column from the original benchmark. Each category is scored from 1-5. Higher is better.
>
> |  | Relevance  | Coherence | Emotional Impact | Originality | Total |
> | :---- | :---: | :---: | :---: | :---: | :---: |
> | Greedy | 4.50 | 3.27 | 2.70 | 2.68 | 13.15 |
> | T=1 | 4.48 | 3.15 | 2.81 | 2.67 | 13.11 |
> | Min-p=0.1 | 4.48 | 3.38 | 2.80 | 2.69 | 13.35 |
> | Human written baseline | 4.50 | 3.81 | 3.03 | 3.2 | 14.54 |
> | Calibrated-TopK | 4.58 | 3.52 | 2.77 | 2.68 | 13.55 |
> | Calibrated-μ | 4.60 | 3.47 | 2.77 | 2.69 | 13.53 |
>
> Thanks for your feedback\! We are happy to discuss more if there are any more concerns.

---

> ### Author Response · Authors · 2025-11-26
>
> Thank you again for your thoughtful review. We wanted to briefly follow up to ask whether our responses about OOD generalization, maj@k improvements and creative writing sufficiently address your questions, or if there is anything else we can clarify.

---

### Official Review · Reviewer_2ZR5 · 2025-11-02

**Soundness:** 4
**Presentation:** 3
**Contribution:** 3
**Rating:** 8
**Confidence:** 4

**Summary:**

This paper re-examines predicting tree search in reasoning models by characterising the difference in uncertainty pertaining to randomness and that due to model uncertainty (aleatoric vs. epistemic uncertainty).  The authors then retrofit the sampling model to differentiate between these two potential sources of uncertainly through their proposed **Greedy-Threshold** sampling, where they calibrate the model's own certainty prediction (as modeled by final-layer logit) against that of the ground truth through Calibrated-$\epsilon$ mapping.

**Strengths:**

* I enjoyed reading this paper.  It is well-structured and presents well-founded arguments for the need for a calibration grid and its derivates that are then used to decide on rollouts.
* Analysis that mid-confidence bins benefit from diverse sampling is intuitive and shown in S2.2.  This is intuitive but needed to be shown in contrast to the prevailing notion that low-confidence requires diverse sampling.
* The paper uses open-weight models Qwen2.5 and Llama, and later the recent GPT-OSS 20B as an advanced LRM.  This aids reproducibility.
* The paper also experiments over various task datasets that are relevant to show cross-task applicability.

**Weaknesses:**

* I wished more information from the Appendix on failure cases (e.g., A.3 and A.6) could make it into the paper proper.  It helps to give a more concrete form with respect to sampling when grounded to an example.
* Same with respect to A.8, the relation to temperature could be given stronger theoretic guidance and derivation.  This part should tie in elegantly, but in the current submission, does not give a sufficiently unified presentation.
* The non-monotonicity of the calibration grids on page 23 deserve a bit of discussion.  The bottom of the grids somewhat reinforce your arguments, but here is is not clear whether sample sparsity also plays a role.

**Questions:**

* Why choose 10 bins?  Could you better fit this as an optimisation parameter where the bin sizes also correlate with transitions in the calibration bin to improve performance?

---

> ### Author Response · Authors · 2025-11-17
> **Theoretical derivation and confidence bin sizes discussion**
>
> We appreciate your in-depth review and nuanced understanding of our motivation and approaches.
>
> * **Regarding the failure case studies being only in the appendix**. We have moved the discussion of failure cases caused by noisy calibration from A.8 into the main body. We show that out-of-domain calibration is generally okay, and still preferable when in-domain data is noisy or sparse.
> * **Theoretical derivation for temperature scaling.** Thanks for the suggestion for stronger theoretical derivation. We have added to Appendix A.8 a calculation for how thresholds should change with temperature. We show that the threshold at new temperature T is simply $\varepsilon_T \approx \varepsilon_1^{1/T}$ compared to at temperature 1, in order to prune a comparable part of the tail of the probability distribution. This is in line with the intuition that truncation thresholds should be lower at lower temperatures.
> * **Non-monotonicity in calibration grids.** Thanks for noticing discrepancies in the calibration grids. The bottom non-monotonicity could be due to a lack of datapoints in low-confidence regimes, so the calibration is noisy with missing values. We have added some discussion about how to deal with noisy calibration, by using out-of-domain (OOD) data in the main body (page 9). We add an additional experiment that uses OOD calibration for coding to show its effectiveness (Table 6).
> * **Design choice of 10 bins.** We chose 10 bins to balance bias and variance. With more bins, lower confidence and higher ranked tokens will receive too few samples and produce noisy correctness estimates for a calibration dataset of a fixed size. On the other hand, with very few bins, we might lose information on the correctness-probability relationship. We avoid tuning for bin size that might result in overfitting depending on the model and dataset. **We provide an additional ablation showing the effect of bin size and and bin width on performance.** We find that a) spacing bins based on quantiles results in a worse linear fits and generally does not perform well, and b) evenly spaced bins with various sizes yield similar performances.
> * Table: Comparison of bins size and bin width on performance of Qwen2.5-0.5B-Instruct on GSM8K. Evenly spaced bins are the default, with the same bin widths. Quantile bin widths are determined by frequency of confidence occurrence. Each bin would contain the same number of samples. A and B are the bias and gradient for linear log-log fit.
>
> | Bins | A | B | maj@8 | maj@16 | maj@32 |
> | :---- | :---: | :---: | :---: | :---: | :---: |
> | 5 even | \-0.574 | 0.791 | 41.2 | 45.4 | 47.6 |
> | 5 quantile | \-2.290 | 0.435 | 33.9 | 34.1 | 34.0 |
> | 10 even | \-0.506 | 0.795 | 40.8 | 44.3 | 47.1 |
> | 10 quantile | \-2.232 | 0.442 | 33.5 | 33.9 | 33.7 |
> | 20 even | \-0.537 | 0.795 | 40.4 | 44.4 | 47.6 |
> | 20 quantile | \-2.214 | 0.444 | 33.5 | 33.6 | 33.6 |
> | 30 even | \-0.500 | 0.802 | 40.6 | 44.5 | 47.1 |
>
>
> Thank you for your detailed comments and constructive suggestions. Please let us know if you have further questions.

---

> > ### Comment · Reviewer_2ZR5 · 2025-11-23
> > **Thank you for your comments**
> >
> > Dear authors, thank you for your rebuttal.  These help me better understand your contribution.
> >
> > I keep my scores as-is as, in my opinion, I feel your submission meet my bar and should be publishable.

---

> > > ### Author Response · Authors · 2025-11-24
> > >
> > > Thank you very much for your thoughtful review, constructive feedback and for supporting the acceptance of our work!

---

### Meta-Review · Area_Chair_Svf7 · 2026-01-05

**Summary:**

The paper analyzes the relation between confidence and correctness. Based on the analysis, the paper proposes new sampling strategies: Greedy-Threshold, Calibrated-TopK, and Calibrated-ε. The reviewers agree that the paper is well-structured and easy to reproduce. But some reviewers raise concerns about marginal performance improvements.

In the rebuttal, the authors emphasize that their methods are learning-free with zero additional inference cost. But it is the same for EDT and ε-sampling. Thus, I consider that the performance concern is not fully addressed.

Although the performance concern remains, the paper identifies an interesting issue with the sampling strategy and proposes a reasonable solution. Thus, following the reviewers' recommendation, I recommend acceptance of the paper.

**Reviewer Concerns:**

Resolved concerns
- Reviewer 2ZR5
  - More explanation on failure cases
  - Theoretical guidance on the relation to temperature
- Reviewer 7oG6
  - Validation is limited to math & reasoning. There is no evaluation for creative & open-ended generation
- Reviewer cpeR
  - Validation on a large Qwen model is required


Remaining concerns
- Reviewer 7oG6
  - Paper relies on calibration, which implies limited generalization without additional cost
  - Improvements are incremental. There is no significant difference from the original big model performance
- Reviewer 2drS
  - Writing issues
- Reviewer cpeR
  - Improvement is marginal

**Reviewer Scores:**

- Reviewer 2ZR5: Would maintain the current score.
- Reviewer 7oG6: Would maintain the current score.
- Reviewer 2drS: Would maintain the current score. The reviewer raises writing and presentation issues, which are hard to address with short comments.
- Reviewer cpeR: Would maintain the current score. Performance concern remains.

---

### Decision · Program_Chairs · 2026-01-26

Accept (Poster)